# Rapid self-heating synthesis of Fe-based nanomaterial catalyst for advanced oxidation

Fengbo Yu [1], Chao Jia [1], Xuan Wu[1], Liming Sun [1], Zhijian Shi [1], Tao Teng[1], Litao Lin[1], Zhelin He [1], Jie Gao [1], Shicheng Zhang [1,2], Liang Wang[3], Shaobin Wang [4] ✉ & Xiangdong Zhu [1,5] ✉

Iron-based catalysts are promising candidates for advanced oxidation process-based wastewater remediation. However, the preparation of these materials often involves complex and energy intensive syntheses. Further, due to the inherent limitations of the preparation conditions, it is challenging to realise the full potential of the catalyst. Herein, we develop an iron-based nanomaterial catalyst via soft carbon assisted flash joule heating (FJH). FJH involves rapid temperature increase, electric shock, and cooling, the process simultaneously transforms a low-grade iron mineral (FeS) and soft carbon into an electron rich nano $Fe^0$/FeS heterostructure embedded in thin-bedded graphene. The process is energy efficient and consumes 34 times less energy than conventional pyrolysis. Density functional theory calculations indicate that the electron delocalization of the FJH-derived heterostructure improves its binding ability with peroxydisulfate via bidentate binuclear model, thereby enhancing ·OH yield for organics mineralization. The Fe-based nanomaterial catalyst exhibits strong catalytic performance over a wide pH range. Similar catalysts can be prepared using other commonly available iron precursors. Finally, we also present a strategy for continuous and automated production of the iron-based nanomaterial catalysts.

The organic contamination of soil and water has seriously threatened human health, especially in developing countries and sparsely populated regions due to incomplete organic pollution treatment[1,2]. Advanced oxidation process (AOP) which produces highly reactive oxygen species (such as hydroxyl radical) by metal material catalyzed oxidants (peroxydisulfate), has been regarded as a promising strategy to deal with the ever-growing problem of organic pollutants[3,4]. Therefore, robust material plays a vital role in addressing the issue of organic pollutants from wastewater through AOP.

Over the past century, Fe-based materials have been one of the most important classes of catalytic materials and have been widely investigated in AOP applications. Conventional thermal treatment in Fe-based materials synthesis is a high temperature for a long-duration process (~hours), leading to metal agglomeration and causing low atom-utilization efficiency[5]. Some investigators also employed

[1]Shanghai Technical Service Platform for Pollution Control and Resource Utilization of Organic Wastes, Shanghai Key Laboratory of Atmospheric Particle Pollution and Prevention (LAP3), Department of Environmental Science and Engineering, Fudan University, 200092 Shanghai, China. [2]Shanghai Institute of Pollution Control and Ecological Security, 200092 Shanghai, China. [3]School of Energy and Power, Jiangsu University of Science and Technology, 212003 Zhenjiang, China. [4]School of Chemical Engineering and Advanced Materials, The University of Adelaide, Adelaide, SA 5005, Australia. [5]National & Local Joint Engineering Laboratory for Municipal Sewage Resource Utilization Technology, Suzhou University of Science and Technology, 215009 Suzhou, China. ✉e-mail: shaobin.wang@adelaide.edu.au; zxdjewett@fudan.edu.cn

defective edges or N, O, and S heteroatoms in a carbon substrate to anchor metals to decrease the agglomeration and provide multiple active sites[6,7]. However, most synthetic methods are tedious processes with high-energy input[8-10]. Significantly, the maximum temperature in conventional thermal treatment may limit the in situ formation of superior components because breaking the chemical bond of some minerals requires a higher preparation temperature[11]. Meanwhile, carbon substrate is often low-quality at such carbonization temperature[12], resulting in poor electron transport ability in AOP[13]. Overall, the further development of robust Fe-based materials is still hampered by inferior catalytic constituents and structure, as well as tedious preparation processes with high-energy input[10,14]. Therefore, solving the aforementioned bottleneck problems would be a breakthrough in reforming Fe-based material.

FJH technique can provide an ultra-high temperature and electric shock by current-induced heating to ~3000 K in seconds and subsequent cooling to room temperature at a cooling rate of $10^5 K s^{-1}$[12,15]. Therefore, FJH could bring some advantages to overcome conventional shortcomings in preparing a superior Fe-based material for AOP: (1) instantaneous ultra-high temperature and electric shock can decompose and crush raw Fe mineral into $Fe^0$ to form a synergistic heterostructure; (2) ultrafast processing actions can effectively inhibit the metal agglomeration to produce highly dispersive metal active sites; (3) the ultra-high temperature and electric shock action can convert the carbon substrate into thin-bedded graphene[12]. These multiple actions in the FJH process are favorable for improving catalytic performances to eliminate recalcitrant organics in AOP. Thus, FJH processing may overcome conventional weaknesses and open up a new avenue to develop a series of more effective Fe-based materials for AOP.

To confirm the hypothesis above, we developed a soft carbon assisted FJH for the synthesis of electron rich nano $Fe^0$/FeS heterostructure embedded in graphene and compared their catalytic performances in AOP with that from conventional pyrolysis. The effects of carbon substrate and FJH power on catalytic degradation of chloramphenicol (CAP) were systematically studied to understand the structure-property relationships. We also investigated various Fe precursors for a general application of FJH and developed automation equipment for large-scale production of a superior Fe-based material in AOP application.

## Results and discussion
### Soft carbon assisted FJH for improving catalytic performance
The schematic diagram for the synthesis of Fe-based material by FJH and conventional pyrolysis is illustrated in Fig. 1a. Before the synthesis of Fe-based material, a soft carbon (hydrochar) was first obtained from hydrothermal liquefaction of biomass (rice straw) and then mixed with FeS powder (a typical low-grade Fe mineral) and carbon black. The mixed mixture was referred to as the raw material (Fe-C-raw). The Fe-based materials from FJH and pyrolysis treatment were denoted as Fe-C-FJH and Fe-C-PY, respectively. The device of FJH was shown in Supplementary Fig. 1a, b, and the reaction process was initiated by applying a voltage (Supplementary Fig. 1c). Sequentially, a current occurred to produce Joule heat, and the sample temperature was significantly elevated. Based on the changes in current and temperature (Fig. 1b), the typical FJH reaction can be divided into two stages (0–3.5 s of an initial stage and 3.5–15 s of a stable stage). In the initial stage, black smoke was produced with slight lighting, indicating that the reaction was started and soft carbon was carbonized (Fig. 1c). After that, with the decreasing resistance of the carbonized sample, the current was sharply increased to produce more Joule heating. Accordingly, sample temperature and lighting intensity were increased due to elevated Joule heating (Supplementary Fig. 2). Finally, this sample could be ultrafast cooled down to room temperature[12,15]. These FJH features, including ultra-high temperature, electrical field, and

ultrafast cooling, can greatly change the sample components and structure. However, in conventional pyrolysis, the low temperature can barely change the structure, and the long-lasting duration lead to metal agglomeration (Supplementary Fig. 3).

In AOP tests, an antibiotic drug, CAP, was degraded via catalytic activation of sodium peroxydisulfate (PDS). Fe-C-FJH showed a superior removal efficiency of up to 94.1%, while Fe-C-raw and Fe-C-PY presented inferior removal efficiencies of 17.0 and 8.78%, respectively (Fig. 1d). In terms of the removal process, a larger $k_{obs}$ (reaction rate constant) value was gained on Fe-C-FJH than that of Fe-C-raw and Fe-C-PY (Supplementary Fig. 4) and its adsorption of CAP on Fe-C-FJH was negligible (Supplementary Fig. 5). In addition, the soft carbon substrate significantly improved CAP removal efficiency (Fig. 1e and Supplementary Fig. 6), because the soft carbon substrate (hydrochar) helps to produce high temperatures to enhance the self-heating of the precursor (Supplementary Fig. 7). Therefore, different soft carbon was further selected as a carbon substrate for FJH processing, and it was found that hydrochar (higher resistance) derived Fe-based material had a better CAP degradation ability compared to pyrochar (lower resistance) derived Fe-based material. The hydrochar can acquire a higher sample power for enhancing the self-heating reaction (Supplementary Fig. 8). Meanwhile, Fe-C-FJH had a higher CAP removal efficiency than the pure Fe catalytic material or composite prepared by conventional methods (Fig. 1f and Supplementary Table 1), which further illustrated the advantages of FJH technology[16-19].

### FJH-induced $Fe^0$/FeS heterostructure
To clearly explain the structure differences of Fe-based materials from different synthesis methods, we checked Fe components and structure. X-ray diffraction (XRD, Supplementary Fig. 9a) showed that FeS, $Fe^0$, and $Fe_3C$ were the main Fe species of Fe-C-FJH, while FeS was the only Fe species of Fe-C-raw and Fe-C-PY. $^{57}$Fe Mössbauer spectra can further effectively differentiate these species and Fe-C-FJH showed the presence of FeS, $Fe_7S_8$, $Fe^0$, $Fe_3C$, and $Fe_2O_3$ at abundances of 26.49, 24.98, 14.94, 29.32, and 4.27%, respectively (Fig. 2a and Supplementary Table 2)[20-23]. $Fe_7S_8$ was the pyrrhotite-4M phase of FeS[20]. FJH induced FeS transformation to $Fe^0$ due to the partial breakage of the Fe-S bond at the ultra-high temperature process (~1800 K). Accordingly, partial $Fe^0$ escaped from FeS and was dissolved in the surrounding carbon atoms and oxidized by oxygen of carbon substrate to form $Fe_3C$ and $Fe_2O_3$, respectively[24-26]. In contrast, Fe-C-raw and Fe-C-PY contained mainly FeS due to the low-temperature process. X-ray photoelectron spectroscopy (XPS) analysis of Fe-C-FJH also confirmed the existence of $Fe^0$ peaks at 706.8 eV, $Fe^{2+}$ peaks at 708.8, 714.8, and 723.2 eV, $Fe^{3+}$ peaks at 711.0 and 724.6 eV (Supplementary Fig. 9c). X-ray absorption near-edge structure (XANES) spectra of Fe-C-FJH showed that the rising edge was situated between Fe foil and FeS, suggesting the Fe atoms with an average oxidation state between $Fe^0$ and $Fe^{2+}$ (Supplementary Fig. 10a)[5]. Moreover, the Fourier-transformed $k^3$-weighted extended X-ray absorption fine structure (EXAFS) (Supplementary Fig. 10b) and its fit spectra (Fig. 2b and Supplementary Table 3) indicated the produced bond of Fe-Fe at the peak of 2.46 Å and radical distance of Fe-S decreased to 2.36 Å after FJH treatment. The S coordination number of Fe in Fe-C-FJH was reduced from 4 to 2, and the Fe coordination number of Fe at 2.46 Å was increased to 2, compared to the FeS standard. It proved the partial breakage of the Fe-S bond and S volatilization in the ultra-high temperature process to produce $Fe^0$ filled into the sulfur position. The wavelet transforms (WT) also indicated the existence of Fe-S and Fe-Fe coordination, compared to that of the standards (Fig. 2c and Supplementary Fig. 11). Due to the breakage of the Fe-S bond and S volatilization, the sulfur morphology and constitution of Fe-C-FJH were also partially converted from $S^{2-}$ and $S_n^{2-}$ to $SO_x$, S-O, and C-S bonds (Supplementary Fig. 12 and Supplementary Table 4)[27,28]. In addition, Fe-C-FJH showed a better CAP degradation efficiency with lower Fe content (Supplementary Fig. 13). These results

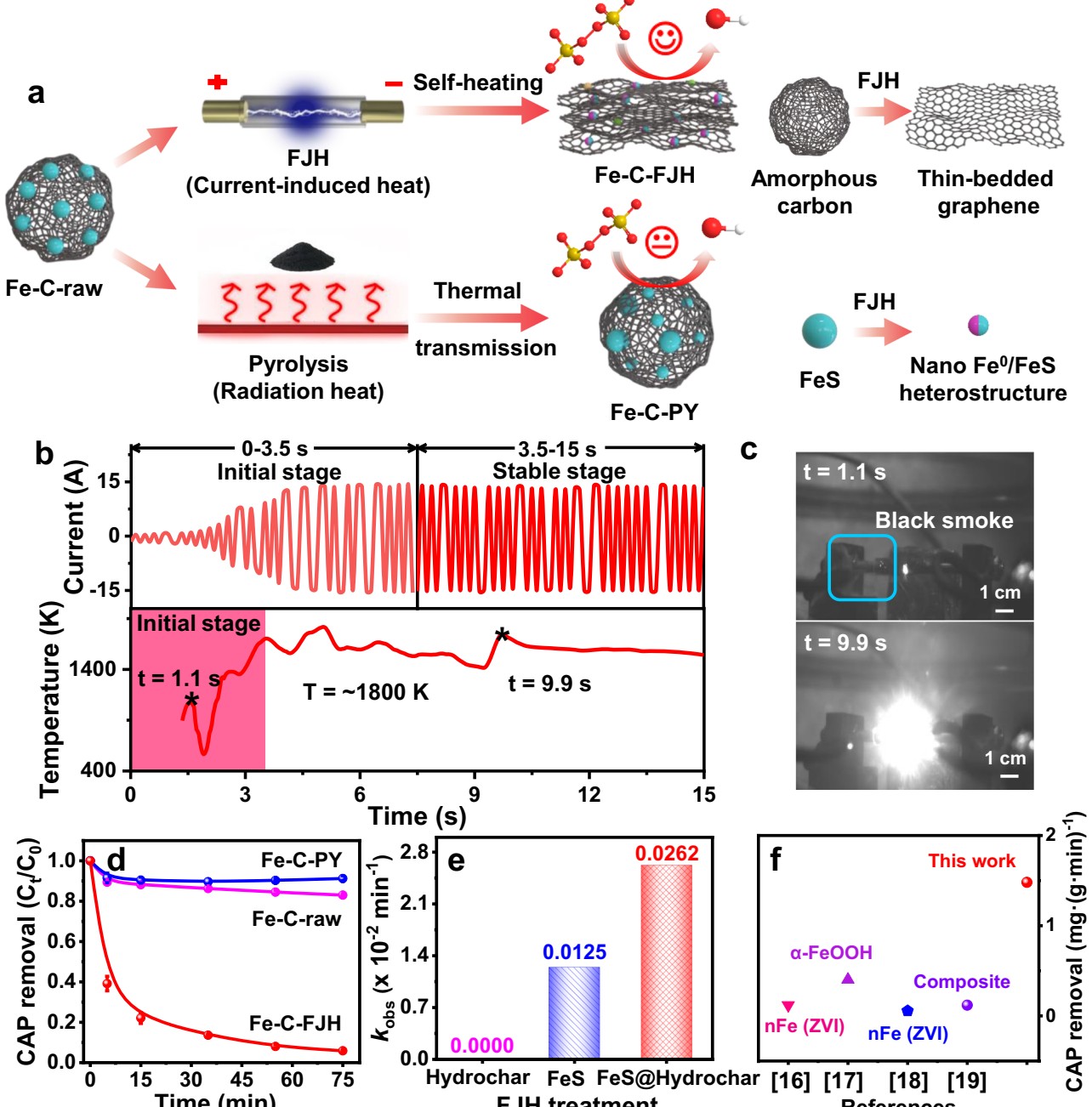

**Fig. 1 | Synthesis of AOP based Fe-based material and their performances in CAP degradation. a** The schematic diagram for the preparation of Fe-based material. After the original raw material (Fe-C-raw) was treated by flash Joule heating (FJH) and pyrolysis technologies and these were regarded as Fe-C-FJH and Fe-C-PY respectively. **b** Current and temperature changes in the FJH process. **c** Lighting intensity changed in the reacted phenomenon recorded by a high-speed camera. **d** CAP degradation profiles by Fe-based material from FJH and pyrolysis. The experiments were repeated twice. **e** Kinetic analysis of catalytic CAP degradation for explaining the role of the carbon substrate. FJH with different precursors independently, hydrochar precursor, FeS precursor, and composite precursor of hydrochar and FeS. **f** Comparison of CAP degradation efficiency from similar AOP reacted systems, such as a pure iron and the iron composite prepared by conventional technologies in the literature. More details were seen in Table S1. Degradation condition: $[CAP]_0 = 60\ mg\ L^{-1}$, [material] = 1000 mg $L^{-1}$, [PDS] = 7 mmol $L^{-1}$, initial $pH_0 = 3.0 \pm 0.2$, temperature = 28 °C.

confirmed that the changed structure by the FJH process have good effects on improving the catalytic performance.

Transmission electron microscopy (TEM) images with energy-dispersive spectroscopy (EDS) elemental mapping clearly showed that Fe and sulfur elements were dispersed uniformly with smaller particle sizes (16.8 nm) (Fig. 2d, e and Supplementary Fig. 14) after FJH treatment. The reason was that ultrafast heating could crush the bulk metal particle and further inhibit metal agglomeration to produce smaller particles due to ultrafast cooling[29]. However, pyrolysis slightly

changed the Fe-C-PY structure due to thermal transmission way for long durations of time. Interestingly, we also found the (112) crystalline lattice of $Fe^0$ and (114) crystalline lattice of FeS to form the nano $Fe^0$/FeS heterostructure by High-angle annular dark-field scanning transmission electron microscopy (HAADF-STEM) image (Fig. 2f). The mapping of electron energy loss spectroscopy (EELS) of STEM preliminarily eliminated the $Fe_2O_3$ component ($Fe^{3+}$) due to the separation of Fe and O elements (Supplementary Fig. 15a–c) and the disappeared O K edge of STEM-EELS in region A and B (Supplementary Fig. 15d–f). In

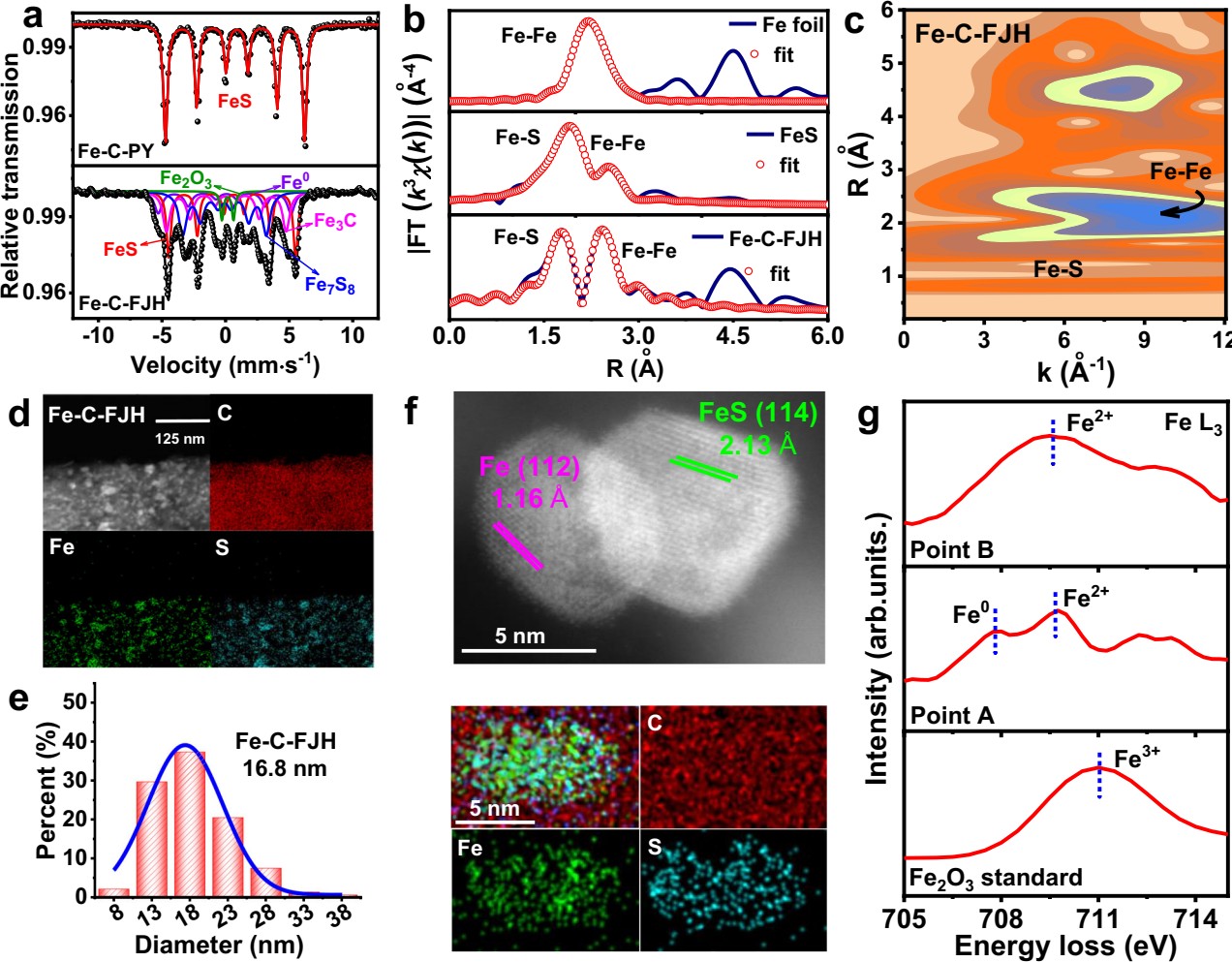

**Fig. 2 | The component and structure analysis of Fe-based material. a** Fe Mössbauer spectra of Fe-C-FJH and Fe-C-PY. **b** EXAFS fitting curves of Fe foil, FeS, and Fe-C-FJH in R space. **c** Wavelet transform analysis of Fe-C-FJH. **d** TEM with EDS elemental mappings of C, Fe, and S elements distribution in Fe-C-FJH. **e** The Fe composite particle size of Fe-C-FJH. **f** HAADF-STEM image and EDS mapping of C, Fe, and S elements distribution in Fe-C-FJH. **g** Fe $L_3$ STEM-EELS data of Fe-C-FJH from point A and point B in Supplementary Fig. 15d.

addition, the Fe L edge of STEM-EELS further indicated the Fe chemical states were $Fe^0$, $Fe^{2+}$ (point A), and $Fe^{2+}$ (point B) (Fig. 2g)[30,31]. The formation reason was that $Fe^0$ was produced from a fast-breaking Fe-S bond due to the decomposition and carbothermic reduction reactions (Supplementary Figs. 16 and 17) and partial $Fe^0$ was sharply solidified near FeS for forming $Fe^0$/FeS heterostructure due to the rapidly cooling rate (up to 541 K s$^{-1}$ in Supplementary Fig. 18)[12]. The other $Fe^0$ escaped from FeS, dissolved in the surrounding carbon atoms, and then oxidized by oxygen of the carbon substrate to form $Fe_3C$ and $Fe_2O_3$. Therefore, all results above indicated that the nano $Fe^0$/FeS heterostructure was synthesized by the Fe-S bond breakage in the FJH process.

**FJH-induced thin-bedded graphene structure**

The fitting of C $1s$ XPS peak (Supplementary Fig. 19a) indicated that the abundance ratio of C=C to C–C had a higher value on Fe-C-FJH, as confirmed by the ratio of sp²/sp³ (Supplementary Table 5). Results implied greater graphitization (Supplementary Fig. 19b) because of carbonization at ultra-high temperature[12,32]. The Raman spectra also confirmed that the intensity of the $G$ peak increased and the $2D$ peak appeared after FJH treatment (Supplementary Fig. 20). In addition, the $I_G/I_D$ value was increased from 0.31 (Fe-C-raw) to 0.87 (Fe-C-FJH), and the $I_{2D}/I_G$ value was 0.81 for Fe-C-FJH (Supplementary Table 5), indicating a high-quality graphene-like structure (thin-bedded graphene)

produced by ultra-high temperature carbonization and electric shock in FJH process[12,32]. The cyclic voltammetry and electrochemical impedance spectroscopy curves (Supplementary Fig. 21) further proved the redox reaction and electron transfer ability of the prepared material[33,34]. The redox peaks of Fe-C-FJH appeared at −0.09 and −0.85 potential (V vs. Ag/AgCl), explaining the Fe redox reaction[35]. Electrochemical impedance spectroscopy indicated that the electron transfer ability of Fe-C-FJH was improved than Fe-C-raw due to the lowest semicircle[36]. The influence of big surface area on electron transfer ability was also eliminated (Supplementary Fig. 22). The surface areas of Fe-C-FJH and Fe-C-PY were produced due to evolving gases (such as $H_2O$, $CO_2$, and $CH_4$[37]) to physically form pores. FJH-derived Fe-based material has a smaller Brunauer−Emmett−Teller (BET) surface area than that of pyrolysis-derived Fe-based material. The reason was that the carbon atom would be rearranged, which led to the closure of micropores and collapse of the wrinkled sheet structure for high graphitization during ultra-high temperature[12,38]. These results indicated thin-bedded graphene produced in FJH process improved the electron transfer ability of Fe-C-FJH for higher reaction efficiency.

**CAP degradation mechanism of Fe-based material**

FJH-induced structures, such as nano $Fe^0$/FeS heterostructure and thin-bedded graphene, may play an important role in PDS activation for CAP degradation. Only $SO_4^{-}$ and ·OH signals were observed in the Fe-C-

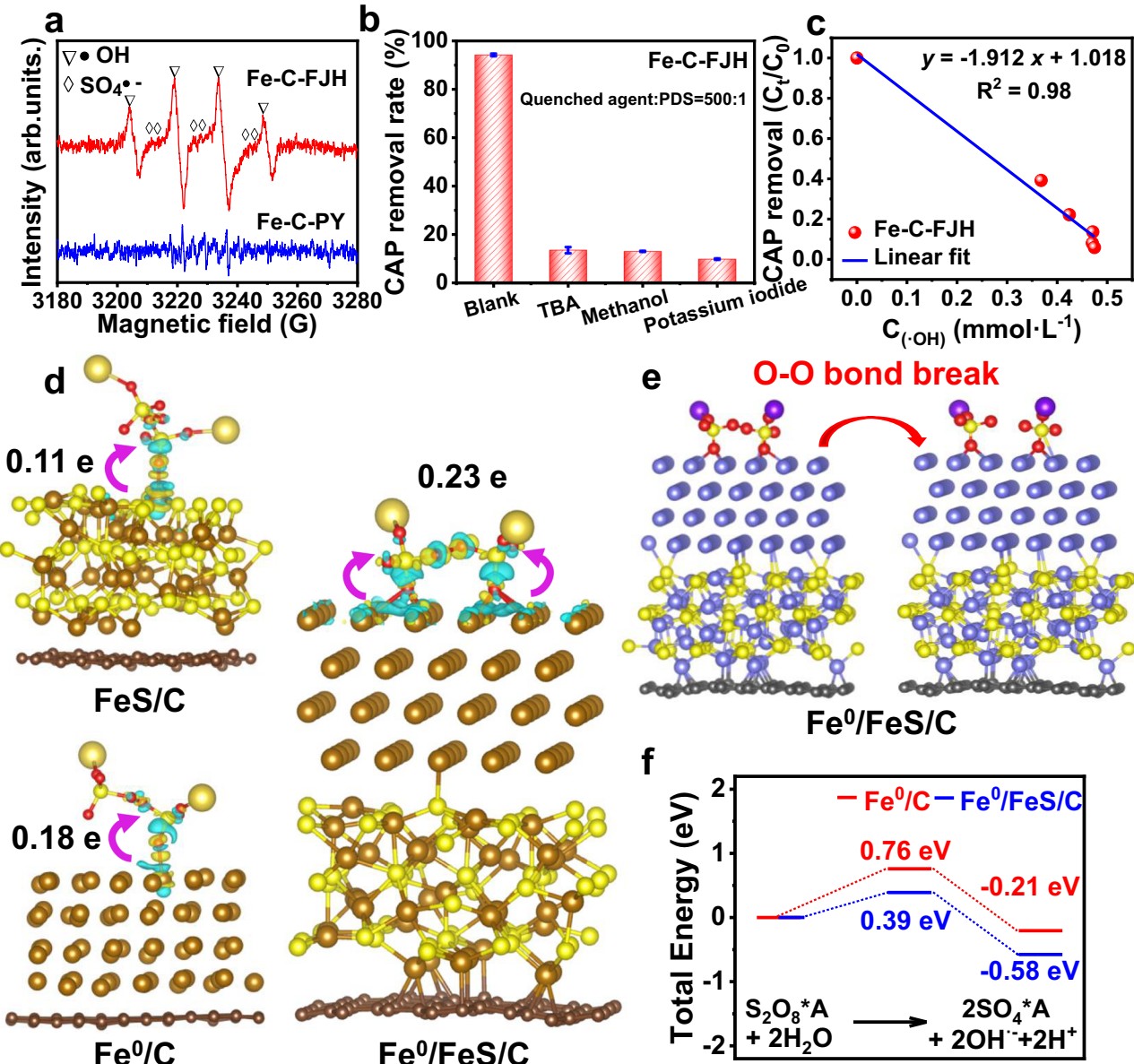

**Fig. 3 | Catalytic mechanism of Fe-based material. a** Electron paramagnetic resonance (EPR) spectra using 5,5-dimethyl-1-pyrroline *N*-oxide (DMPO) as a trapping reagent for •OH and SO$_4$•⁻. **b** The inhibition efficiency of Tert-butanol (TBA), methanol, and potassium iodide for •OH quenching reagent, •OH and SO$_4$•⁻ quenching reagent, and surface-free radical quenching reagent, respectively. The experiments were repeated twice. **c** The line fit of •OH concentration and CAP removal. **d** The electron density difference of PDS adsorption on different Fe-based materials. FeS, Fe⁰, the composite of FeS and Fe⁰ embedded in graphene were regarded as FeS/C, Fe⁰/C, and Fe⁰/FeS/C, respectively. **e** The O-O bond breakage process of PDS on Fe⁰/FeS/C. **f** The free energies for PDS activation by Fe⁰/C and Fe⁰/FeS/C on the Fe sites. S$_2$O$_8$*A and 2SO$_4$*A represented the absorbed configuration.

FJH/PDS system, while not in the Fe-C-PY/PDS system (Fig. 3a and Supplementary Fig. 23). The intensity of the SO$_4$•⁻ signal was weak because of the conversion of SO$_4$•⁻ to •OH[5,10]. Radical scavenger experiments further indicated that the reactive oxygen species was •OH and the reaction occurred on the surface of Fe-based material (Fig. 3b). The CAP degradation efficiency could be positively correlated with the •OH and dissolved Fe³⁺ concentrations (Fig. 3c and Supplementary Fig. 24). The •OH and Fe³⁺ concentrations were positively correlated. Because of dissolved Fe³⁺, the solution pH of Fe-C-FJH was also slightly decreased. The degradation of CAP due to Fe³⁺ salts and the dissolved Fe ions obtained from Fe-C-FJH/PDS/CAP was negligible (Supplementary Fig. 24e, f). In addition, only negligible CAP degradation was observed in the presence of commercially sourced Fe$_3$C and Fe$_2$O$_3$ (Supplementary Fig. 25). Fe⁰/FeS heterostructure was

also the main contributor to CAP degradation due to the 66.41% mass content of total Fe (Supplementary Table 2)[4,39]. Therefore, the results above indicated the active components were Fe⁰/FeS heterostructure.

The degradation intermediate products from CAP/PDS/Fe-C-FJH system at different degradation times were shown in Supplementary Figs. 26 and 27. Consistent with the observations from previous reports[40–45], the primary degradation reactions of CAP degradation included oxidation, dehydrogenation, cleavage of the amide group, and hydroxylation for producing small molecular components. Importantly, these tiny molecular components were further mineralized into inorganic ions (such as NO$_3$⁻ and Cl⁻) by dehalogenation, denitration, and oxidation reaction (Supplementary Table 6). Therefore, the proposed degradation pathway is shown in Supplementary Fig. 28.

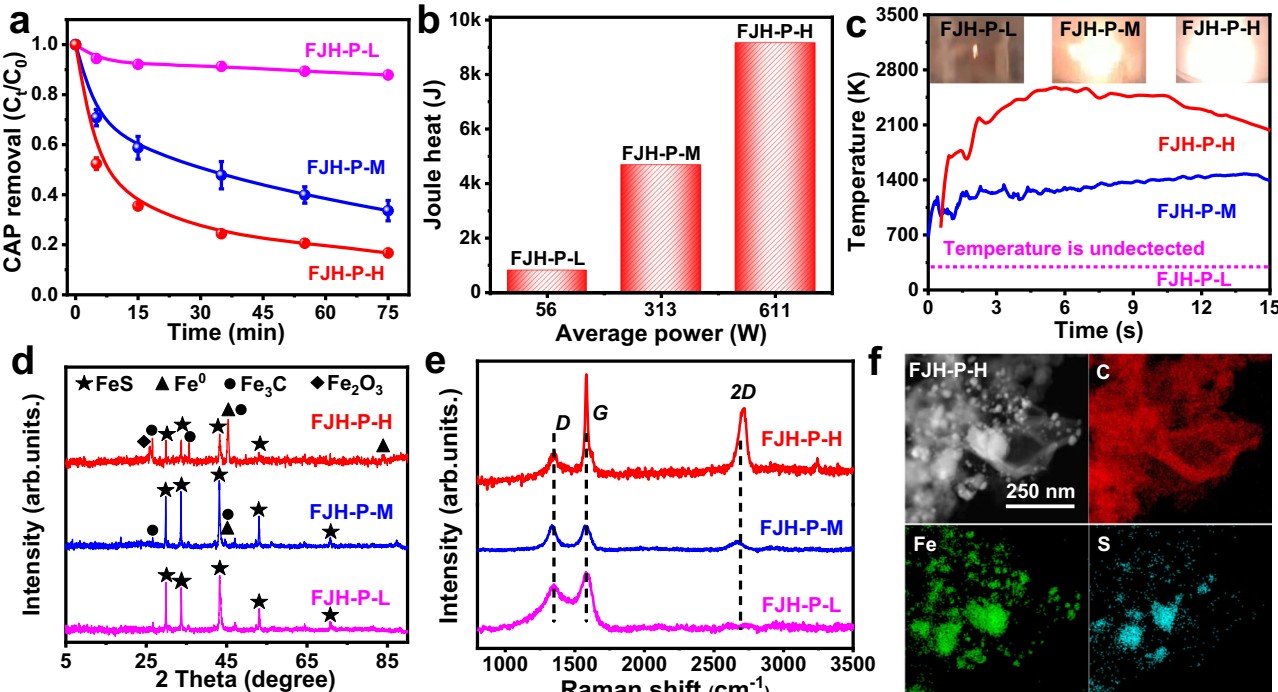

**Fig. 4 | Analysis of FJH power on catalytic performance. a** The CAP removal of Fe-based material prepared at different FJH power. According to the FJH power, Fe-based materials were regarded as FJH-L, FJH-M, and FJH-H respectively. The experiments were repeated twice. Degradation condition: $[CAP]_0 = 60\ mg\ L^{-1}$, $[material] = 1000\ mg\ L^{-1}$, $[PDS] = 7\ mmol\ L^{-1}$, initial $pH_0 = 3.0 \pm 0.2$, temperature $= 28\ ^\circ C$. **b** Average power and Joule heat of prepared processes for Fe-based material. The power was acquired using the multiplication of recorded voltage and current. Joule heat was acquired by multiplying power and time. **c** Temperatures were recorded in FJH processes. **d** XRD analysis of Fe-based material prepared at different FJH power. **e** Raman analysis of Fe-based material prepared at different FJH power. **f** EDS elemental mappings of C, Fe, and S element distributions in FJH-P-H.

## Density functional theory (DFT) calculations

Furthermore, we theoretically unraveled the importance of thin-bedded graphene in the PDS activation process. The adsorption systems of single FeS and FeS/C composite with PDS verified that the PDS-binding affinity of FeS/C was raised from −2.026 to −2.126 eV, and O-O bond length of PDS was extended from 1.527 to 1.530 Å (Supplementary Figs. 29 and 30a, b). As evidenced by the electron density difference, electron transmission ability was strengthened by thin-bedded graphene, suggesting the higher adsorption ability of PDS on the FeS/C (Fig. 3d and Supplementary Fig. 30c)[46,47].

The DFT calculation further explained the catalytic mechanism of nano $Fe^0/FeS$ heterostructure composite. We structured two species structure models ($Fe^0/FeS/C$ and $FeS/Fe^0/C$) about $Fe^0/FeS$ heterostructure embedded in graphene. The adsorption energy on $Fe^0/FeS/C$ and $FeS/Fe^0/C$ were -4.169 eV and -4.135 eV, respectively. The O-O bond length of PDS was 1.622 Å and 1.619 Å, respectively. The result indicated that compared to the adsorption systems of FeS/C or $Fe^0/C$ with PDS, $Fe^0/FeS$ heterostructure easily adsorbed PDS and broke O-O bond to produce radicals due to the increase of electron transmission ability. The electron transmission ability of $Fe^0/FeS/C$ and $FeS/Fe^0/C$ were notably higher than FeS/C and $Fe^0/C$ (Fig. 3d and Supplementary Fig. 30c) because of the electron delocalization effect that more electrons can be transferred[46,48]. Electron delocalization triggered bidentate binuclear model formation, providing two passageways for electron transformation between the Fe atom and PDS. The difference of $Fe^0/FeS/C$ and $FeS/Fe^0/C$ was negligible in adsorption energy, O-O bond of PDS, and electron transmission ability. All results confirmed that existent electron rich nano $Fe^0/FeS$ heterostructure greatly improved CAP degradation.

In addition, the breakage of O-O bond was a necessary vital process in this catalytic reaction. Therefore, the PDS activation reactions on $Fe^0/C$ and $Fe^0/FeS/C$ from the beginning to the end were selected and constructed to investigate corresponding intermediate processes (Fig. 3e and Supplementary Fig. 31a). And we interestingly found that, compared to $Fe^0/C$, electron rich $Fe^0/FeS/C$ has a lower energy barrier for the O-O bond breakage and the catalytic reaction has a lower Gibbs free energy (−0.58 eV) (Fig. 3f and Supplementary Fig. 31b, c). The electron delocalization effect made it spontaneous and easier for electron rich $Fe^0/FeS/C$ to activate PDS.

## FJH power effects on the catalytic performance

The effects of FJH power on the catalytic performance toward CAP degradation were further studied. It was found that the CAP degradation efficiency was improved sharply with increasing power and Joule heat (Fig. 4a, b and Supplementary Figs. 32 and 33). The high power can induce a strong FJH reaction, as suggested by the higher sample brightness and temperature (Fig. 4c). However, lower power could not provide adequate Joule heat to break the Fe-S bond and exfoliate the carbonized substrate to form nano $Fe^0/FeS$ heterostructure and thin-bedded graphene. Therefore, the degradation efficiency of FJH-P-L was poor. According to the XRD patterns (Fig. 4d), $Fe^0$, $Fe_3C$, and $Fe_2O_3$ were produced, and the peak intensities of $Fe^0$ increased with increasing power. In addition, the degree of graphitization also increased (Fig. 4e). The $I_G/I_D$ increased from 0.47 (FJH-P-L) to 1.21 (FJH-P-H), and $I_{2D}/I_G$ of FJH-P-H reached 1.43 (Supplementary Table 7), indicating the formation of more thin-bedded graphene[12]. EDS elemental maps of TEM images showed an inconsistent distribution of S and Fe elements, which suggested the existence of $Fe^0$ and FeS. Meanwhile, the disappeared oxygen and the $Fe^0$ (point A) and $Fe^{2+}$ (point B) chemical states from STEM-EELS further verified the existence of $Fe^0$ and FeS (Fig. 4f and Supplementary Fig. 34)[30,31]. In the catalytic tests, the relationship between $Fe^{3+}$ concentration and CAP degradation efficiency ($C_t/C_0$) was fitted (Supplementary Fig. 35). These results were consistent with the above result of Fe-C-FJH. And

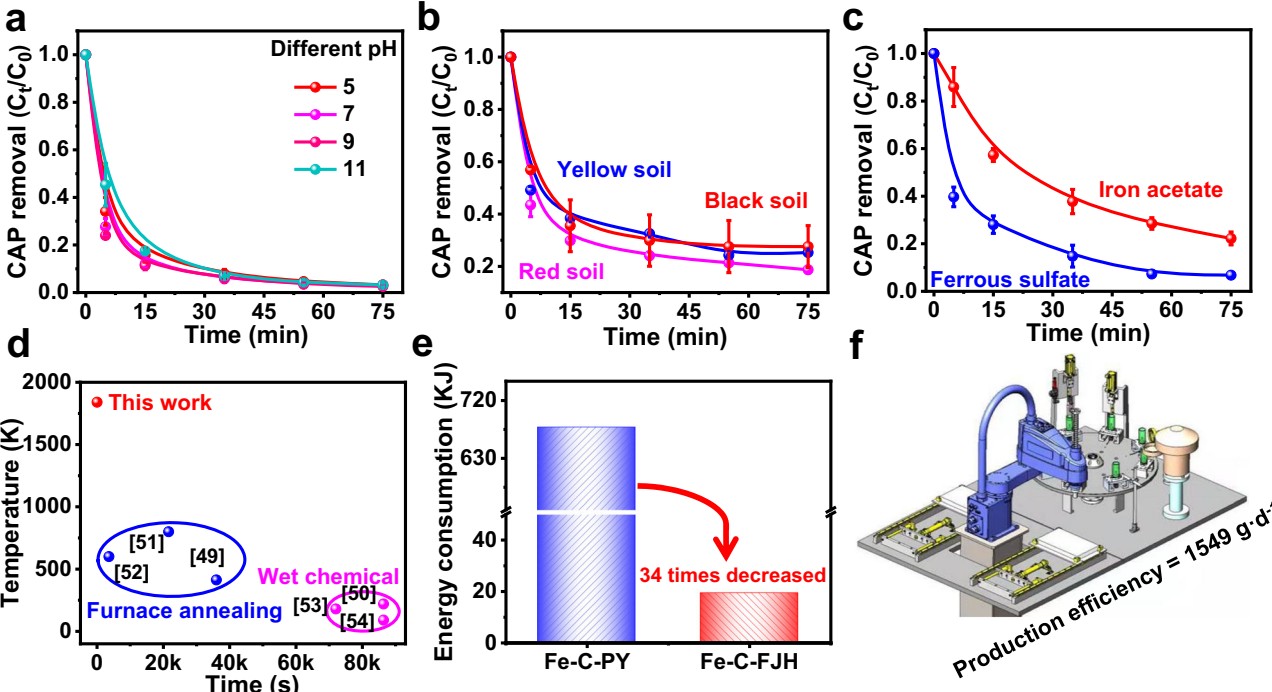

**Fig. 5 | Adaptability and large-scale production potentiality of FJH technology.** **a** CAP removal of Fe-C-FJH at different pH ranges. **b** CAP removal of Fe-C-FJH at different soil systems. **c** The applicability of FJH for different Fe salts precursors toward CAP degradation. The experiments were repeated twice (**a**–**c**). **d** Summary of nanomaterials synthesis temperatures and duration time by conventional technologies in the literature. **e** The energy consumption comparison between FJH and pyrolysis in terms of the experiment process. **f** Large-scale fabrication of Fe-based material via automation device. Degradation condition: (**a**) $[CAP]_0 = 60\,mg\,L^{-1}$, $[material] = 1000\,mg\,L^{-1}$, $[PDS] = 7\,mmol\,L^{-1}$, initial $pH_0 = 5$–11, (**b**) $[CAP]_0 = 60\,mg\,L^{-1}$, $[material] = 2000\,mg\,L^{-1}$, $[soil] = 100\,g\,L^{-1}$, $[PDS] = 7\,mmol\,L^{-1}$, initial $pH_0 = 3.0 \pm 0.2$, (**c**) $[CAP]_0 = 60\,mg\,L^{-1}$, $[material] = 1000\,mg\,L^{-1}$, $[PDS] = 7\,mmol\,L^{-1}$, initial $pH_0 = 3.0 \pm 0.2$.

the higher power was beneficial for forming nano $Fe^0$/FeS heterostructure and thin-bedded graphene and these structured materials can activate PDS to produce more active radicals for CAP degradation.

## FJH adaptability for preparation of Fe-based material

To investigate the Fe-based material for an industrial application, we first studied the Fe-based material in various environments. The material exhibited significant CAP degradation efficiency over a wide pH range (Fig. 5a and Supplementary Fig. 36a). The material also can be applied to the remediation of different types of soil (properties shown in Supplementary Table 8). The material showed an excellent CAP degradation efficiency in various soil types and the removal effect of only soil without Fe-based material can be negligible (Fig. 5b and Supplementary Fig. 36b). The optimal mass ratio of FeS to hydrochar was 0.6 for CAP degradation in aqueous solutions (Supplementary Fig. 36c). In addition, FJH-derived material from different Fe precursors (Ferrous sulfate and iron acetate) always presented superior CAP degradation efficiency than those prepared by conventional pyrolysis (Fig. 5c and Supplementary Fig. 37), suggesting that FJH as a robust technology had a general implication for improving the structure of both low-grade Fe mineral and soluble Fe salt in AOP application.

In addition, we compared the synthesis temperature and time with those of conventional techniques reported in literature[49–54]. The FJH had a higher temperature and shorter duration (15 s) for Fe-based material preparation (Fig. 5d). The energy consumption of conventional pyrolysis was 34 times as high as that of FJH (Fig. 5e). We are developing a continuous device (Fig. 5f and Supplementary Fig. 38a) for the automated production of Fe-based material. This device included a loading and collecting area, a robot arm (Supplementary Fig. 38b), and a reacted technics area (Supplementary Fig. 38c). The robot arm can continuously transfer raw material from the loading area to the reacted technics area. After the reaction ends, the robot arm moves Fe-based material by FJH from the reacted technics area to the collecting area. A complete process needs only 20 s and the production of Fe-based material will be 1549 g day$^{-1}$ by one designed device according to the prospective design (Supplementary Table 9). The device could be further magnified to improve sharply the production efficiency. Undoubtedly, FJH could be developed for large-scale production of superior Fe-based material with low energy consumption for industrialization by automation equipment.

In summary, we report a soft carbon assisted FJH technology by a current-induced self-heating to fabricate a superior Fe-based material and its effectiveness for organic pollution removal in a wide pH range and different soil types application. FJH technique prepared an electron rich nano $Fe^0$/FeS heterostructure embedded in graphene by the synchronously ultra-high temperature (-1800 K) and electric shock. A carbon substrate with a higher soft degree (high abundance of sp$^3$ *C-C* bonds) would generate catalytic material to exhibit a better CAP degradation efficiency due to the high derived power and Joule heat. DFT calculations indicated that FJH-induced electron rich nano $Fe^0$/FeS heterostructure embedded in graphene could form a bidentate binuclear model with PDS to increase the electron transformation ability and O-O bond length for the recalcitrant organic removal. Moreover, FJH technology has demonstrated its effectiveness in accommodating various conditions (like different Fe salts and carbon substrate precursors) for instantaneous preparation, and the energy consumption is 34 times lower than that of conventional pyrolysis. Therefore, FJH can be used to synthesize Fe-based material and has great promise in industrialization for large-scale AOP applications.

## Methods

### Preparation of Fe-based material

**Pretreatment.** A soft carbon substrate (hydrochar) was first obtained from hydrothermal liquefaction of biomass (rice straw). Rice straw

(15.0 g) and distilled water (210 mL) were loaded into a 500 mL autoclave. The autoclave was heated to 270 °C for 60 min at a stirring speed of 300 rpm. After the reaction was completed, the hydrochar was acquired by filtration of the mixed product. Finally, the solid was dried as hydrochar. Then, hydrochar power (1.00 g) and FeS powder (as typical low-grade Fe mineral, 0.60 g) were uniformity mixed in deionized water by oscillation and then dried in a vacuum drying oven for further use. Finally, carbon black (regarded as a conductive additive) at 10 wt% was mixed with such dried solid for FJH treatment. This mixed sample was denoted as Fe-C-raw.

**FJH treatment.** To synthesize a robust Fe-based material, Fe-C-raw was treated by a homemade FJH device. Briefly, Fe-C-raw (0.12 g) was put into a quartz tube and compressed with copper electrodes to minimize the sample resistance to ~200 Ω. Then, a mild vacuum (more than 38 Kpa) was kept in the reaction process to avoid sample oxidation. Finally, Fe-based material (Fe-C-FJH, FJH-P-L, FJH-P-M, FJH-P-H) was synthesized by controlling FJH parameters (Supplementary Table 10), including operation voltage, external resistance, and reaction time.

To explain the reacted phenomenon of the FJH process, the voltage and current were recorded by an oscilloscope (RIGOL, DS1104Z Plus). The temperature profile was further collected using an infrared spectrometer and fitting blackbody radiation. Lighting intensity was recorded by a high-speed camera and further treated by a Hue, Saturation, and Value color model and Matrix Laboratory (MATLAB) software.

**Conventional pyrolysis treatment.** Fe-C-raw was added to the porcelain boat and put into a tubular furnace. Then, the tubular furnace was heated (5 K min$^{-1}$) up to 973 K for 90 min under a 100 mL min$^{-1}$ N$_2$ flow. After the pyrolysis process, the sample was collected as Fe-C-PY.

## Data availability

All the data supporting the findings of this study are available in the article and the Supplementary Information files. Source data are provided with this paper.

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

## Acknowledgements

This work was supported by the National Natural Science Foundation of China (No. 22276040 to X.Z.), the National and Local Joint Engineering Laboratory for Municipal Sewage Resource Utilization Technology, Suzhou University of Science and Technology (No. 2021KF03 to X.Z.), and Future Food (Bai Ma) Research Institute announced the leading project (JBGS-2021-007 to X.Z.). The authors also thank scientific compass for help in DFT analysis.

## Author contributions

F.Y. and X.Z. conceived and designed the experiments, F.Y., C.J., X.W., L.S., Z.S., T.T., L.L., Z.H., J.G., L.W., and X.Z. performed the experiments; F.Y., S.Z., S.W., and X.Z. analyzed the data; F.Y., S.Z., S.W., and X.Z. wrote the paper.

## Competing interests

The authors declare no competing interests.
