## [Peer Review File · Nature Communications]

Rapid self-heating synthesis of Fe-based nanomaterial catalyst for advanced oxidationREVIEWER COMMENTS

Reviewer #1 (Remarks to the Author):

Attached please find.

Reviewer #2 (Remarks to the Author):

In this work, the authors reported the synthesis of FeO/FeS/C heterostructure embedded in graphene for catalytic degradation of organic contaminants in wastewater by advanced oxidation. With regard to the catalyst aspect, I am not expertise in this specific catalytic reaction, so I cannot judge if the presented catalytic performance is novel enough over other literature methods. On the synthesis by the flash Joule heating process, many critical elements are missing. These noted below should be addressed or clarified.

1. The author ascribed the conversion of FeS to Fe(0) as a high-temperature decomposition process. However, because of the existence of carbon in the FJH process, it is more likely to be a carbothermic reduction process. If the carbothermic reduction temperature of FeS is within the range of a conventional furnace, the advantages of FJH synthesis conversion over that of a furnace synthesis is not pronounced. In FJH, carbon often acts as the reductant—much like the classical method of reducing metal salts using carbon and a flame.

2. The key claim of this manuscript is the synthesis of FeO/FeS/C heterostructures. However, there seems to be a lack of evidence for the formation of the FeO/FeS heterostructure. On Line 160, the authors mentioned “HAADF-STEM with EDS also explained the FeO was existed in surface of FeS”. I am confused on how such conclusion is reached. The EDS maps only show the elemental distribution and never tell the chemical state of the elements. If the authors want to obtain such a conclusion, they should provide EELS to distinguish the chemical state. Without EELS, all is speculation.

3. Similarly, on Line 236, the authors claim, “TEM images with EDS elemental mapping clearly showed the existed FeO and FeS due to the inconsistent distribution of S and Fe elements”. The EDS maps are insufficient to draw this conclusion since it could be Fe₂O₃, Fe₃C, etc.

4. The Mossbauer spectra show the presence of FeS, Fe₇S₈, FeO, Fe₃C, and Fe₂O₃ at abundances of 26.49%, 24.98%, 14.94%, 29.32%, and 4.27%, suggesting complicated compositions of Fe species in the product. However, for the catalytic mechanism explanation, the authors only considered the heterostructure of FeO/FeS. The discussion of other Fe species to the catalytic performance was totally overlooked.

5. In Supplementary Fig. 18, the Fe-C-FJH obtained from FJH has a much smaller BET surface area than that of Fe-C-PY obtained from pyrolysis. Could the author explain this?

6. In the last part of the manuscript, the authors proposed a design for the automated production. Has that already been built or is it just a design? If it is built, it would be interesting to see how it scales (engineering issues) and the sample uniformity when scaling up.

7. The English should be greatly improved to avoid many painful grammatical errors.

Reviewer #3 (Remarks to the Author):

The article titled “Instantaneous self-heating boost Fe-based material for advanced oxidation decontamination” has technical advancement. However, modification is required if accepted for publication.

1. Abstract. Better to not use any abbreviation without the full form
2. Introduction: too many fancy sentences; please write them properly.
 - a. The sentence starting with Luckily
 - b. This unique “two birds with one stone” feature
3. What do you mean by self-heating in figure 1a?
4. Flash joule heating demonstrated, and directly the biomass can be used for FJH, here, authors used hydrothermal liquefaction; why?
5. Table S2 seems misleading; please put a line after FE-C-PY
6. DFT analysis has provided more details; however, more discussion should be added in SI on how authors came to know about the possible input structure for the DFT analysis for their characterization studies.
7. Chloramphenicol used in the study is too high? The author should provide the reason for the same.
8. The authors should provide the full chromatograph for the intermediates as Fig S21 has partial. By this, it is difficult to understand how the authors have reached the degradation pathway.
9. It seems 80% removal within 15 minutes and not much effect of pH. It should be checked the mechanism of degradation. Is it adsorption followed by degradation? Have the authors attempted to understand the role of adsorption? Please provide a chromatograph for all important degradation steps.
10. The authors tried to put too many things in the manuscript; however, details are missing for many and should be added.

Reviewer #4 (Remarks to the Author):

The study by Yu et al, describes the FJH synthesis of a Fe based catalyst for advanced oxidation process pollutant degradation. Despite the fact that FJH is now well known, the authors use the technique to develop a superior AOP material, which is thoroughly described and complete.

The authors need to thoroughly improve the english throughout the manuscript to improve clarity.

The manuscript deals with the preparation of a novel Fe-based material (Fe-C-FJH, Fe⁰/FeS heterostructure embedded in thin-bedded graphene) by FJH method and its performance of efficient peroxydisulfate (PDS) activation for CAP degradation. The ultra-high temperature, electric shock, and rapid cooling functions in FJH reaction can decompose and crush raw Fe mineral into Fe⁰ to form a synergistic heterostructure, effectively inhibit the metal agglomeration to produce highly dispersive metal active sites, and convert the carbon substrate into thin-bedded grapheme. The resulting Fe-C-FJH showed an enhanced OH yield for CAP mineralization due to the increased binding ability with peroxydisulfate via bidentate binuclear model. The topic is of significance in environmental catalysis, and the results seem interesting. However, I do not recommend its acceptance for publication in Nature Communication due to the weakness in environmental application and discussion of PDS activation. The specific comments are as following.

1. It was noticed that the CAP degradation processes were performed at initial pH₀=3.0±0.2, iron species (Fe⁰/FeS embedded in graphene) of the Fe-C-FJH would inevitably dissolved and released into the solution. As stability is the key of catalysis, I do not think the Fe-C-FJH is potential for real environmental water remediation. In addition, the CAP degradation efficiency in various soil types with the Fe-C-FJH was also evaluated at pH 3, such an acid soil is impracticable in soil site remediation. The experiment design is suggested to considering real background of environmental issue.
2. As shown in Fig. 3c and Fig. S20, the CAP degradation efficiency was positively correlated with the OH and dissolved Fe³⁺ concentrations, and the OH and Fe³⁺ concentrations was positively correlated. Thus the control experiment of homogeneous Fe³⁺/PDS should be considered and discussed. Did the released Fe³⁺ contribute to the PDS activation and CAP degradation?

Reviewer #1 (Remarks to the Author):

The manuscript deals with the preparation of a novel Fe-based material (Fe-C-FJH, Fe⁰/FeS heterostructure embedded in thin-bedded graphene) by FJH method and its performance of efficient peroxydisulfate (PDS) activation for CAP degradation. The ultra-high temperature, electric shock, and rapid cooling functions in FJH reaction can decompose and crush raw Fe mineral into Fe⁰ to form a synergistic heterostructure, effectively inhibit the metal agglomeration to produce highly dispersive metal active sites, and convert the carbon substrate into thin-bedded grapheme. The resulting Fe-C-FJH showed an enhanced ·OH yield for CAP mineralization due to the increased binding ability with peroxydisulfate via bidentate binuclear model. The topic is of significance in environmental catalysis, and the results seem interesting. However, I do not recommend its acceptance for publication in Nature Communication due to the weakness in environmental application and discussion of PDS activation. The specific comments are as following.

A: We are very appreciated for that you pointed these questions about the weakness in environmental application and discussion of PDS activation. In order to solve these concerns, we added many experiments considering real background (such as homogeneous Fe³⁺/PDS experiments) and explained the mechanisms in depth. We hope this revised manuscript will be satisfactory and can solve these concerns. If you have further any questions, please, we are so happy to answer to make this manuscript accurate and strong for publication.

Q1. *It was noticed that the CAP degradation processes were performed at initial pH₀ = 3.0 ± 0.2, iron species (Fe⁰/FeS embedded in graphene) of the Fe-C-FJH would inevitably dissolved and released into the solution. As stability is the key of catalysis, I do not think the Fe-C-FJH is potential for real environmental water remediation. In addition, the CAP*

degradation efficiency in various soil types with the Fe-C-FJH was also evaluated at pH 3, such an acid soil is impracticable in soil site remediation. The experiment design is suggested to considering real background of environmental issue.

A: Thank you for your valuable suggestions. We added some experiments for further clarifying the potential of material in practical applications.

(1) We firstly explained the problem of iron ion dissolved and released into the solution. The catalytic mechanism of Fe-C-FJH was that Fe⁰/FeS heterostructure embedded in graphene activated PDS into ·OH for CAP degradation and Fe³⁺ was also produced. Therefore, Fe species of Fe-C-FJH were inevitably dissolved and released into the solution. However, compared with other literatures^{1,2,3}, the dissolved efficiency of Fe ions owing to CAP removal, was lower in this study (**Fig. 1**). Results indicated Fe-based material prepared by FJH technology achieved superior removal efficiency and lead to a lower dissolved Fe ions level. In addition, Fe-based materials prepared by FJH have a great CAP removal efficiency for soil restoration. Therefore Fe-based materials could be used in soil application and need not to be retrieved. Meanwhile, the dissolved Fe ions were environmentally friendly for rehabilitated soil.

Fig. 1 Dissolved Fe ion efficiency of per gram CAP removal from Fe-C-FJH and other Fe-based material reported in literatures.

(2) We are sorry for your misunderstanding about the acidification problem of soil owing to that the Fe-C-FJH sample was applied in soil restoration. Fe-based material prepared by FJH technology can be applied in wide pH range (3~11) and have superior CAP removal efficiency (**Fig. 5a in this revised manuscript**). Therefore, the pH at 3 was not necessary for soil restoration. In addition, we also measured the pH of dried black soil was 6.22 from the system of Fe-C-FJH/soil/PDS/CA at pH 3. Results indicated dried black soil can be recovered from the acid environment. The CAP removal efficiency in various soil types with the Fe-C-FJH was also evaluated without regulating pH. Results indicated CAP removal efficiency of Fe-C-FJH was high and CAP removal efficiency without PDS or Fe-C-FJH was negligible (**Fig. 2**). Therefore, all above results inferred that Fe-C-FJH has a great CAP removal efficiency in various soil types at different pH for soil site remediation.

Fig. 2 a-c, CAP removal at different soil system with (a) Fe-C-FJH/PDS/CAP, (b) Fe-C-FJH/CAP, and (c) PDS/CAP. Degradation condition: [CAP]₀ = 60 mg/L, [material] = 2000 mg/L (If needed), [PDS] = 7 mmol/L (If needed), [soil] = 100 g/L, temperature = 28 °C.

The revised parts in abstract and introduction were as follows for emphasizing the application of Fe-based material in soil restoration and wastewater treatment.

In abstract, “water scarcity” and “wastewater” were substituted with “organic pollution”.

In introduction, the revised part was as follows.

The organic contamination of soil and water has seriously threatened human health, especially in the developing countries and sparsely populated regions due to incomplete organic pollution treatment^{4, 5}. Advanced oxidation process (AOP) which produces highly

reactive oxygen species (such as hydroxyl radical) by metal material catalyzed oxidants (peroxydisulfate, PDS), have been regarded as a promising strategy to deal with the ever-growing problem of organic pollutants^{6, 7}. Therefore, robust material plays a vital role in addressing the issue of organic pollution wastewater through AOP.

Q2. As shown in Fig. 3c and Fig. S20, the CAP degradation efficiency was positively correlated with the $\cdot\text{OH}$ and dissolved Fe^{3+} concentrations, and the $\cdot\text{OH}$ and Fe^{3+} concentrations were positively correlated. Thus, the control experiment of homogeneous Fe^{3+} /PDS should be considered and discussed. Did the released Fe^{3+} contribute to the PDS activation and CAP degradation?

A: To explain the contribution of Fe^{3+} or dissolved Fe ions on CAP removal, we added these control experiments of homogeneous Fe^{3+} /PDS/CAP and dissolved Fe ions solution from the finished system of Fe-C-FJH/PDS/CAP. Results found that the contribution of Fe^{3+} at high concentration from FeCl_3 (1000 mg/L) was very low due to the inefficient for activating PDS (**Fig. 3a**)⁸. The reacted constant of Fe^{3+} with PDS was only $6.6 \times 10^{-2} \text{ M}^{-1} \text{ S}^{-1}$. And the CAP removal of dissolved Fe ions was only 7.2% due to the new adding PDS (**Fig. 3b**). Therefore, the CAP removal was due to that Fe-C-FJH sample activated PDS into $\cdot\text{OH}$ for CAP degradation. Fe^{3+} was produced due to CAP degradation, not contributed to CAP degradation.

Fig. 3 a-b, a The effect of Fe^{3+} on CAP removal, and CAP removal rate at 75 min was put in the inset image. Degradation condition: $[CAP]_0 = 60$ mg/L, $[FeCl_3] = 1000$ mg/L (if need), $[PDS] = 7$ mmol/L (if need), initial $pH_0 = 3.0 \pm 0.2$, temperature = 28 °C. **b** CAP removal of dissolved Fe ions solution from the finished system of Fe-C-FJH/PDS/CAP at 75 min by adding new PDS and CAP components.

CAP removal data by Fe^{3+} and dissolved Fe ions from the system of Fe-C-FJH/PDS/CAP was added in supporting materials as **Supplementary Fig. 21 e and f**. And the explanation of this added **Supplementary Fig. 21 e and f** was as follows.

Because of dissolved Fe^{3+} , the solution pH of Fe-C-FJH was also slightly decreased. The CAP removal of Fe^{3+} and dissolved Fe ions solution from the system of CAP/PDS/Fe-C-FJH could be negligible (Fig. S21e and f). In addition, the CAP removal of commercial Fe_3C and Fe_2O_3 was bad (Fig. S22). Therefore, we inferred the active components could be Fe^0/FeS heterostructure. According to the content of Fe species, the content of Fe^0/FeS heterostructure from FeS and Fe^0 was 66.41% of total Fe, which also indicated that Fe^0/FeS heterostructure was the main contribution for CAP degradation^{7,9}.

Reviewer #2 (Remarks to the Author):

In this work, the authors reported the synthesis of Fe⁰/FeS/C heterostructure embedded in graphene for catalytic degradation of organic contaminants in wastewater by advanced oxidation. With regard to the catalyst aspect, I am not expertise in this specific catalytic reaction, so I cannot judge if the presented catalytic performance is novel enough over other literature methods. On the synthesis by the flash Joule heating process, many critical elements are missing. These noted below should be addressed or clarified.

A: Thank you for great suggestions and comments. We checked and added some key experiments for improving this manuscript according to your suggestions and comments.

Q1. *The author ascribed the conversion of FeS to Fe(0) as a high-temperature decomposition process. However, because of the existence of carbon in the FJH process, it is more likely to be a carbothermic reduction process. If the carbothermic reduction temperature of FeS is within the range of a conventional furnace, the advantages of FJH synthesis conversion over that of a furnace synthesis is not pronounced. In FJH, carbon often acts as the reductant—much like the classical method of reducing metal salts using carbon and a flame.*

A: As you said, carbon was often acted as the reductant in pyrolysis process. However, FeS cannot be reduced by carbothermic reduction due to following reasons.

(1) We added the experiment by increasing pyrolysis temperature (1173 K) which was adequate-high temperature in conventional pyrolysis process. XRD result indicated Fe⁰ was not produced and the CAP removal was also inferior (**Fig. 4**).

(2) In addition, the thermal decomposition temperature of FeS was 1373 K by theoretical analysis. In FJH process, the temperature can be ~1800 K and FeS easily decompose at this temperature¹⁰.

(3) Pyrolysis in Fe-based materials synthesis is a high-temperature for long-duration process (~hours), leading to metal agglomeration due to high surface energy and causing

low atom-utilization efficiency. However, FJH technology not only possessed an ultra-high temperature advantage, but also electric shock and rapid cooling functions to acquire Fe⁰/FeS heterostructure with smaller size metal particle and thin-bedded graphene, which could improve sharply the catalytic performance.

Fig. 4 a-b. (a) XRD analysis and (b) CAP removal profiles of Fe-C-PY prepared by pyrolysis at 1173 K for 90 min. Degradation condition: [CAP]₀ = 60 mg/L, [material] = 1000 mg/L, [PDS] = 7 mmol/L, initial pH₀ = 3.0±0.2, temperature = 28 °C.

Q2. *The key claim of this manuscript is the synthesis of Fe⁰/FeS/C heterostructures. However, there seems to be a lack of evidence for the formation of the Fe⁰/FeS heterostructure. On Line 160, the authors mentioned “HAADF-STEM with EDS also explained the Fe⁰ was existed in surface of FeS”. I am confused on how such conclusion is reached. The EDS maps only show the elemental distribution and never tell the chemical state of the elements. If the authors want to obtain such a conclusion, they should provide EELS to distinguish the chemical state. Without EELS, all is speculation.*

A: Thank you for your suggestions, and we added the STEM-EELS data for further explaining the formation of Fe⁰/FeS heterostructure. The mapping of STEM-EELS preliminarily eliminated the Fe₂O₃ component (Fe³⁺) due to the separation of Fe and O element (**Fig. 5a-c**). And this oxygen was existed in the carbon substrate as indirectly proved by the disappeared O K edge of STEM-EELS in region A and B (**Fig. 5d-g**).

Fig. 5 a-c, HAADF-STEM with EDS mapping of Fe and O elements distribution in Fe-C-FJH. **d-g**, Fe L₃ STEM-EELS data from point A and point B.

In addition, the Fe L edge of STEM-EELS further indicated the Fe chemical states were Fe⁰ and Fe²⁺ (point A) and Fe²⁺ (point B)^{11, 12}. The formation reason was that Fe⁰ was produced from fast breaking Fe-S bond and partial Fe⁰ was sharply solidified near to FeS for forming Fe⁰/FeS heterostructure due to the rapidly cooling at a cooling rate of 10⁵ K/s¹³. The other Fe⁰ escaped from FeS and was dissolved in the surrounding carbon atoms and oxidized by oxygen of carbon substrate to form Fe₃C and Fe₂O₃. Therefore, all results (STEM-EDS and STEM EELS) proved the formation of the Fe⁰/FeS heterostructure.

The added data was added as **Supplementary Fig. 15 and Fig 2g**. The explanation part was revised in line 160-171 in the manuscript.

Q3. Similarly, on Line 236, the authors claim, “TEM images with EDS elemental mapping clearly showed the existed Fe⁰ and FeS due to the inconsistent distribution of S and Fe elements”. The EDS maps are insufficient to draw this conclusion since it could be Fe₂O₃, Fe₃C, etc.

A: I agree with your comment that only EDS maps are insufficient to draw the existed Fe⁰

and FeS. Therefore, we added the STEM-EELS data for further explaining the existed Fe⁰ and FeS.

XRD result firstly indicated the existence of Fe⁰, FeS, Fe₃C, and Fe₂O₃. According to the TEM-EDS mapping, the Fe and S elements were existed and the site of Fe and S elements were not coincided. Therefore, results of XRD and TEM-EDS may infer the partial existing of Fe⁰ and FeS. Meanwhile, the disappeared oxygen, the Fe⁰ (point A) and Fe²⁺ (point B) chemical states from STEM EELS further verified the existing of Fe⁰ and FeS (**Fig. 6**)^{11, 12}.

The STEM-EELS results were added as **Supplementary Fig. 31** in the revised supporting materials. And the corresponding explanation was revised in line 257 to line 261 in the revised manuscript.

Fig. 6. Fe L₃ STEM-EELS data of FJH-P-H sample from point A and point B.

Q4. The Mossbauer spectra show the presence of FeS, Fe₇S₈, Fe⁰, Fe₃C, and Fe₂O₃ at abundances of 26.49%, 24.98%, 14.94%, 29.32%, and 4.27%, suggesting complicated compositions of Fe species in the product. However, for the catalytic mechanism

explanation, the authors only considered the heterostructure of Fe⁰/FeS. The discussion of other Fe species to the catalytic performance was totally overlooked.

A: Thank you for your suggestion. We added some experiments for further explaining clearly the catalytic mechanism of Fe species.

(1) Firstly, Fe⁰ was produced from fast breaking Fe-S bond at ultra-high temperature and partial Fe⁰ was sharply solidified near to FeS for forming Fe⁰/FeS heterostructure due to the rapidly cooling at a cooling rate of 10⁵ K/s¹³. The other Fe⁰ escaped from FeS and was dissolved in the surrounding carbon atoms and oxidized by oxygen of carbon substrate to form Fe₃C and Fe₂O₃. Fe₇S₈ was the pyrrhotite-4M phase of FeS¹⁴.

(2) Then, we also added the CAP degradation experiments of commercial Fe₃C and Fe₂O₃. Results found the CAP removal of commercial Fe₃C and Fe₂O₃ with PDS was bad (**Fig. 7**). Meanwhile, the results of homogeneous Fe³⁺ and dissolved Fe ions solution from the finished system of Fe-C-FJH/PDS/CAP at 75 min on CAP removal, found that the contribution of Fe ions was very low due to the inefficient for activating PDS (**Fig. 3**)⁸. The reacted constant of Fe³⁺ with PDS was only 6.6*10⁻² M⁻¹S⁻¹.

(3) The results of DFT calculations also indicated the Fe⁰/FeS heterostructure embedded in graphene had a great activation performance of PDS than single components embedded in graphene.

(4) The content of Fe⁰/FeS heterostructure from FeS and Fe⁰ was also 66.41% of total Fe, and the other content of Fe species was low.

All in all, all above results can infer the Fe⁰/FeS heterostructure was the main activated sites for CAP degradation.

Fig. 7 The CAP removal of commercial Fe₃C and Fe₂O₃. Degradation condition: [CAP]₀ = 60 mg/L, [PDS] = 7 mmol/L (if need), initial pH₀ = 3.0±0.2, temperature = 28 °C. The additive amount of commercial Fe₃C and Fe₂O₃ was the same with the corresponding Fe content of Fe-C-FJH according to the ICP and Fe Mössbauer spectra results.

The CAP degradation experiments of commercial Fe₃C and Fe₂O₃ was added as **Supplementary Fig. 22**, and the information of commercial Fe₃C and Fe₂O₃ was added in supporting material. The corresponding explanation was added in the revised manuscript.

The CAP removal of Fe³⁺ and dissolved Fe ions solution from the system of CAP/PDS/Fe-C-FJH could be negligible (Fig. S21e and f). In addition, the CAP removal of commercial Fe₃C and Fe₂O₃ was bad (Fig. S22). Therefore, we inferred the active components could be Fe⁰/FeS heterostructure. According to the content of Fe species, the content of Fe⁰/FeS heterostructure from FeS and Fe⁰ was 66.41% of total Fe, which also indicated that Fe⁰/FeS heterostructure was the main contribution for CAP degradation^{7, 9}.

Q5. In Supplementary Fig. 18, the Fe-C-FJH obtained from FJH has a much smaller BET surface area than that of Fe-C-PY obtained from pyrolysis. Could the author explain this?

A: The BET surface areas of Fe-C-FJH and Fe-C-PY were produced due to evolving

gases (such as H₂O, CO₂, CH₄¹⁵) to physically form pores. The evolving gases were derived from this decomposition of hydrochar in the ultra-high temperature. While FJH derived Fe-based material has a much smaller BET surface area than that of pyrolysis derived Fe-based material, the reason was that carbon atom would be rearranged which led to closure of micropores and collapse of the wrinkled sheet structure for high graphitization during ultra-high temperature^{13, 16}. The high graphitization of Fe-C-FJH can also be proven by the Raman spectrum (**Supplementary Table 5 and Fig. 16 and 17 in the supporting materials**), while that of Fe-C-PY was lower. Therefore, the BET surface areas of Fe-C-PY obtained from pyrolysis was higher than that of Fe-C-FJH.

The added explanation was as follows in the revised manuscript.

The influence of big surface area on electron transfer ability was also eliminated (Fig. S19). The surface areas of Fe-C-FJH and Fe-C-PY were produced due to evolving gases (such as H₂O, CO₂, CH₄¹⁵) to physically form pores. While FJH derived Fe-based material has a much smaller BET surface area than that of pyrolysis derived Fe-based material, the reason was that carbon atom would be rearranged which led to closure of micropores and collapse of the wrinkled sheet structure for high graphitization during ultra-high temperature^{13, 16}. These results indicated thin-bedded graphene produced in FJH process improved the electron transfer ability of Fe-C-FJH for higher reaction efficiency.

Q6. In the last part of the manuscript, the authors proposed a design for the automated production. Has that already been built or is it just a design? If it is built, it would be interesting to see how it scales (engineering issues) and the sample uniformity when scaling up.

A: We have been building this automated production for scaling up. This preliminary synthesis processes and automation device for large-scale preparation were shown in **Fig. 8**. All automatic processes were controlled by Programmable Logic Controller (PLC) platform. The synthesis processes usually included six steps for acquiring catalytic

material (**Fig. 8a**). The practical picture was shown in **Fig. 8b** and **Fig. 9**. But it needs more time to further complete from design to application. After finishing this automation device for large-scale preparation, the sample uniformity or catalytic performance will also be evaluated in our future research.

Fig. 8 a-b, The schematic route and principle of automated production for scaling up.

Fig. 9 The equipment of automated production for scaling up of studied material.

Q7. *The English should be greatly improved to avoid many painful grammatical errors.*

A: Thank you for your suggestion. We made language polishing by a native speaker from editage (a professional editing service company).

Reviewer #3 (Remarks to the Author):

The article titled “Instantaneous self-heating boost Fe-based material for advanced oxidation decontamination” has technical advancement. However, modification is required if accepted for publication.

A: Thank you so much for your suggestions and comments. We revised this manuscript according to your suggestions and comments.

Q1. *Abstract. Better to not use any abbreviation without the full form.*

A: We appreciate your suggestion. In abstract, I added the full form before the abbreviation for the first appearance.

This revision was as follows,

Advanced oxidation materials shed light on addressing organic pollution while are hampered by the inferior catalytic performance and tedious prepared processes. Herein, we innovatively developed a soft carbon assisted current self-heating technique for instantaneous fabrication (15 s) of a robust Fe-based material. Ultra-high temperature (~1800 K), electric shock, and rapid cooling functions in flash Joule heating (FJH) reaction can simultaneously reform Fe mineral and carbon substrate. This multifunctional feature of FJH synthesized electron-rich nano Fe⁰/FeS heterostructure embedded in thin-bedded graphene ensuing excellent catalytic performance. Density functional theory calculations indicated electron delocalization of FJH derived heterostructure improved binding ability with peroxydisulfate via bidentate binuclear model, finally enhancing ·OH yield for recalcitrant organics mineralization. Importantly, various Fe precursors can be used to prepare superior material via FJH with 34 times lower energy input than synthesis by conventional pyrolysis. Generally, FJH can be developed into an automation device for large-scale application in organic pollution treatment.

Q2. *Introduction: to many fancy sentences; please write them properly.*

a. The sentence starting with Luckily

b. This unique “two birds with one stone” feature

A: As suggested by you, we have also changed these fancy sentences throughout the revised manuscript. We have removed the words “Luckily” at the starting sentence, and substituted “FJH multifunctional feature” for “this unique ‘two birds with one stone’ feature”.

These revisions were as follows,

(1) Flash Joule heating (FJH) technique can provide an ultra-high temperature and electric shock by a current-induced heating to ~3000 K in seconds and subsequent cooling to room temperature at a cooling rate of 10^5 K/s^{11, 14}.

(2) This multifunctional feature of FJH synthesized electron-rich nano Fe⁰/FeS heterostructure embedded in thin-bedded graphene ensuing excellent catalytic performance.

Q3. *What do you mean by self-heating in figure 1a?*

A: The greatest difference between self-heating and pyrolysis is the way of heating sources. Self-heating in this manuscript means that current directly passthrough the sample and further instantaneously (in seconds) induced the sample to produce Joule heating and ultra-high temperature (~1800 K), and temperature was subsequent cooling to room temperature at a cooling rate of 10^5 K/s to change the material structures. Contrarily, the heating-up way of sample in the pyrolysis process was heating transmission (few hours) and sample was heated into high temperature (~1273 K). In addition, current cannot passthrough the sample in pyrolysis process. Therefore, pyrolysis has many limitations in heating rate, temperature ranges, and cooling rate, resulting inferior material structures.

Q4. *Flash joule heating demonstrated, and directly the biomass can be used for FJH,*

here, authors used hydrothermal liquefaction; why?

A: We agree with your comment on that biomass can be used for FJH. We found that hydrochar and biomass as precursor have the same catalytic efficiency. But in terms of biomass as precursor, FJH processes will produce many gases and tar, which may be easier to break reaction quartz tube and hinder the working of FJH, especially in this automated production for scaling up. Hydrochar as precursor from pre-carbonized biomass can decrease the gases and tar emission in the FJH processes, and has a great environmental effectiveness.

Q5. Table S2 seems misleading; please put a line after FE-C-PY.

A: Thank you for your suggestion. I added the line in **Table 1** for clearly reading. This revision was as **Supplementary Table 2** in this supporting material.

Table 1 | Fe components of the Fe-based material prepared by FJH and conventional pyrolysis, detected by ^{57}Fe Mössbauer spectroscopy.

Sample	Site	Hr (KOe)	IS (mm/s)	QS (mm/s)	$\Gamma/2$ (mm/s)	Content (%)
Fe-C-PY	FeS	310.6	0.76	0.14	0.18	100
	FeS	310.12	0.59	-0.15	0.21	26.49
	Fe ₇ S ₈	292.55	-0.01	0.14	0.33	24.98
Fe-C-FJH	Fe ⁰	324.44	0.00	-0.15	0.36	14.94
	Fe ₃ C	204.25	-0.08	-0.01	0.37	29.32
	Fe ₂ O ₃	-	0.17	0.91	0.16	4.27

Q6. DFT analysis has provided more details; however, more discussion should be added in SI on how authors came to know about the possible input structure for the DFT analysis for their characterization studies.

A: Thank you for your suggestion.

(1) We firstly explained the possible input structure by analyzing the characteristics of

material for the DFT analysis. XRD and ^{57}Fe Mössbauer spectra results indicated the existence of FeS and Fe^0 . STEM-EDS and STEM-EELS further proved the formation of the Fe^0/FeS heterostructure. The formation reason was that Fe^0 was produced from fast breaking Fe-S bond and partial Fe^0 was sharply solidified near to FeS for forming Fe^0/FeS heterostructure due to the rapidly cooling at a cooling rate of 10^5 K/s^{13} . Meanwhile, the Fe^0/FeS heterostructure was embedded in thin-bedded graphene which was testified by the high value of I_{2D}/I_G from Raman, C1s of XPS, and HRTEM.

(2) In addition, the Fe^0/FeS heterostructure embedded in thin-bedded graphene was main contribution for improving CAP degradation. In order to explain the superior PDS activation performance of Fe^0/FeS heterostructure embedded in thin-bedded graphene, we compared with FeS embedded in thin-bedded graphene (FeS/C) and Fe^0 embedded in thin-bedded graphene (Fe^0/C). The PDS adsorption energy, O-O bond length of PDS, and electron density difference were acquired for evaluating catalytic performances. In addition, we further explained the role of thin-bedded graphene in activating PDS by structuring the only FeS (FeS without graphene) and FeS embedded in thin-bedded graphene (FeS/C) respectively.

(3) Meanwhile, the reduction of PDS on material to produce reactive free radical was testified, which produced the Na_2SO_4 and $\cdot\text{OH}$ due to the O-O bond breakage of $\text{Na}_2\text{S}_2\text{O}_8^{17}$. Therefore, the PDS activation reactions on Fe^0/C and $\text{Fe}^0/\text{FeS}/\text{C}$ from the begin to the end were constructed to evaluate the energy barrier for the O-O bond breakage and Gibbs free energy for the whole reaction process.

The explanation of input structure was added in density functional theory calculation part in supporting material.

Q7. Chloramphenicol used in the study is too high? The author should provide the reason for the same.

A: We agree with your comment on that chloramphenicol used in the study is too high

than practical wastewater. But we found the chloramphenicol (10 ppm) was completely removed at reacted time of 5 min (**Fig. 10**). Therefore, in order to further illustrate this superior catalytic performance of Fe-C-FJH and distinguish the catalytic difference of different materials, we improve the chloramphenicol concentration in the study.

Fig. 10 The CAP degradation of Fe-C-FJH. Degradation condition: [CAP]₀ = 10 mg/L, [material] = 1000 mg/L, [PDS] = 7 mmol/L, initial pH₀ = 3.0±0.2, temperature = 28 °C.

Q8. *The authors should provide the full chromatograph for the intermediates as Fig S21 has partial. By this, it is difficult to understand how the authors have reached the degradation pathway.*

A: Thank you for your suggestion. According to your suggestion, we added the full chromatograph of GC-MS and HPLC/TOFMS and some experiments for analyzing the intermediates at different degradation times. Results of GC-MS found seven intermediate components from the system of CAP/PDS/Fe-C-FJH at different degraded time, and the peak area of CAP components was decreased with increasing degraded time due to the CAP degradation (**Fig. 11 and Table 2**).

Fig. 11 Intermediate products of CAP degradation by Fe-C-FJH were measured from GC-MS. Degradation condition: $[CAP]_0 = 300$ mg/L, $[material] = 2000$ mg/L, $[PDS] = 14$ mmol/L, initial $pH_0 = 3.0 \pm 0.2$.

Table 2 The peak area of intermediate products of CAP degradation by Fe-C-FJH were measured from GC-MS.

Time (min)	Components	Area		
		5 min	15 min	55 min
4.701	2,2-dichloro-N-(3-methylbutyl) acetamide (C ₇ H ₁₃ Cl ₂ NO)	1843217	188049	998019
5.857	3-Hydroxy-4-nitrobenzaldehyde (C ₇ H ₅ NO ₄)	-	-	57378
7.094	4-Nitrophenol (C ₆ H ₅ NO ₃)	-	-	108204
7.318	4-Nitrobenzoic acid (C ₇ H ₅ NO ₄)	-	-	770185
9.137	3,5-dinitro-,2-(4-nitrophenyl)-2-oxoethyl ester benzoic acid (C ₁₅ H ₉ N ₃ O ₉)	449738	186864	1399161
12.596	2,2-dichloro-N-(1-(4-nitrobenzoyl) vinyl) acetamide (C ₁₁ H ₈ Cl ₂ N ₂ O ₄)	3102696	2557426	4540784
13.163	4-nitro-Benzaldehyde (C ₇ H ₅ NO ₃)	4412269	7909688	
16.161	Chloramphenicol (C ₁₁ H ₁₂ Cl ₂ N ₂ O ₅)	6691069	1046553	824258

In addition, we further analyzed intermediate products from the system of CAP/PDS/Fe-C-FJH at different degraded time by HPLC/TOF-MS. According to GC-MS results and literatures^{1, 18, 19, 20}, the m/z ratio of 122.0266, 138.0213, 166.0173, 197.0228, and 178.0372 and the corresponding structure were analyzed (**Fig. 12**). In order to explain accurately CAP degradation pathway, we revised the degradation pathway which was illustrated in **Fig. 13**. The main pathways of CAP degradation included possibly dehydration, cleavage of the amide group, and oxidation for producing small molecular components. Importantly, these small molecular components were further mineralized into inorganic ions (such as NO₃⁻ and Cl⁻) by dehalogenation, denitration, and oxidation reaction^{1, 18, 19, 20}.

The full chromatograph of GC-MS and HPLC/TOFMS were added as **Supplementary Fig. 23 and 24**. The possible degraded pathway of CAP was added as **Supplementary Fig. 25**. And the corresponding explanation was revised in line 212 to line 218 in the revised manuscript.

Fig. 12 Intermediate products of CAP degradation by the system of CAP/PDS/Fe-C-FJH were measured from HPLC/TOF-MS. Degradation condition: $[\text{CAP}]_0 = 300 \text{ mg/L}$ (if need), $[\text{material}] = 2000 \text{ mg/L}$ (if need), $[\text{PDS}] = 14 \text{ mmol/L}$ (if need), initial $\text{pH}_0 = 3.0 \pm 0.2$.

Fig. 13 Proposed CAP degradation pathways from the system of CAP/PDS/Fe-C-FJH

Q9. It seems 80% removal within 15 minutes and not much effect of pH. It should be checked the mechanism of degradation. Is it adsorption followed by degradation? Have the authors attempted to understand the role of adsorption? Please provide a chromatograph for all important degradation steps.

A: Thank you for your suggestions. We had considered the CAP adsorption of Fe-C-FJH and found the adsorption was low (**Supplementary Fig. 5**). To further address these concerns of adsorption, we also have performed additional adsorption experiments. Results indicated the adsorption ability of Fe-C-FJH at different pH and material (FJH-P-L, FJH-P-M, FJH-P-H) on CAP was low (**Fig. 14**). Therefore, the CAP removal was mainly degraded by Fe-based material.

CAP adsorption of Fe-based materials was added in supporting materials as **Supplementary Fig. 29 c and Supplementary 33a**.

Fig. 14 a-b, The CAP adsorption of (a) Fe-C-FJH at different pH and (b) different Fe-based materials. Degradation condition: [CAP]₀ = 60 mg/L, [material] = 1000 mg/L, initial pH₀ = 3.0±0.2 (if need), temperature = 28 °C.

According to your suggestion, we also added the full chromatograph of GC-MS and HPLC/TOFMS and some experiments for analyzing the intermediates at different degradation times. Results of GC-MS found seven intermediate components from the system of CAP/PDS/Fe-C-FJH at different degraded time (**Fig. 11 and Table 2**).

According to GC-MS, HPLC/TOFMS results and literatures^{1, 18, 19, 20}, the m/z ratio of 122.0266, 138.0213, 166.0173, 197.0228, and 178.0372 and the corresponding structure were analyzed (**Fig. 12**). In order to explain accurately CAP degradation pathway, we revised the degradation pathway which was illustrated in **Fig. 13**. The main pathways of CAP degradation included possibly dehydration, cleavage of the amide group, and oxidation for producing small molecular components. Importantly, these small molecular components were further mineralized into inorganic ions (such as NO₃⁻ and Cl⁻) by dehalogenation, denitration, and oxidation reaction^{1, 18, 19, 20}.

***Q10.** The authors tried to put too many things in the manuscript; however, details are missing for many and should be added.*

A: Thank you for your great comments. According your comments, we checked this manuscript and added more details in this revised manuscript. Meanwhile, we added necessary experiments to explain clearly the catalytic mechanism of Fe-based material. These main details added in this revised manuscript were follows.

(1) We added these control experiments of homogeneous Fe³⁺/PDS/CAP and dissolved Fe ions solution from the system of Fe-C-FJH/PDS/CAP to explain the Fe³⁺ or dissolved Fe ions contribution on CAP removal. Results found that the contribution of high concentration Fe³⁺ from FeCl₃ was very low due to the inefficient for activating PDS. CAP removal of Fe³⁺ and dissolved Fe solution from the system of Fe-C-FJH/PDS/CAP was added in supporting materials as **Supplementary Fig. 21e and f**.

(2) We added these STEM-EELS data for further explaining the formation of the Fe⁰/FeS heterostructure. The added data of STEM-EELS was added as **Supplementary Fig. 15 and Fig 2g** in this revised manuscript.

The revised part was as follows in the revised manuscript.

Interesting, we also found the (112) crystalline lattice of Fe⁰ and (114) crystalline lattice of FeS to form the nano Fe⁰/FeS heterostructure by High-angle annular dark-field scanning

transmission electron microscopy (HAADF-STEM) image (Fig. 2f). HAADF-STEM with EDS may infer the Fe⁰ was existed in surface of FeS. The mapping of electron energy loss spectroscopy (EELS) of STEM preliminarily eliminated the Fe₂O₃ component (Fe³⁺) due to the separation of Fe and O elements (Fig. S15a-c) and the disappeared O K edge of STEM-EELS in region A and B (Fig. S15d-f). In addition, the Fe L edge of STEM-EELS further indicated the Fe chemical states were Fe⁰, Fe²⁺ (point A) and Fe²⁺ (point B) (Fig. 2g)^{11, 12}. The formation reason was that Fe⁰ was produced from fast breaking Fe-S bond and partial Fe⁰ was sharply solidified near to FeS for forming Fe⁰/FeS heterostructure due to the rapidly cooling at a cooling rate of 10⁵ K/s¹³. The other Fe⁰ escaped from FeS and was dissolved in the surrounding carbon atoms and oxidized by oxygen of carbon substrate to form Fe₃C and Fe₂O₃. Therefore, all results above indicated that the nano Fe⁰/FeS heterostructure was synthesized by decomposition and crush of FeS in the FJH process.

(3) We added some explanations in supporting material to know about the possible input structure for the DFT analysis for their characterization studies. The explanation of input structure was added in density functional theory calculation part in supporting material.

(4) We further provided additional chromatograph of GC-MS and HPLC/TOF-MS, and analyzed clearly these degradation pathways. The full chromatograph of GC-MS and HPLC/TOFMS were added as **Supplementary Fig. 23 and 24**. The possible degraded pathway of CAP was added as **Supplementary Fig. 25**. And the corresponding explanation was revised in line 212 to line 218 in the revised manuscript.

(5) To eliminate the adsorption of Fe-C-FJH, we also have performed additional adsorption experiments. CAP adsorption of Fe-based materials was added in supporting materials as **Supplementary Fig. 29 c and Supplementary 33a**.

Reviewer #4 (Remarks to the Author):

The study by Yu et al, describes the FJH synthesis of a Fe based catalyst for advanced oxidation process pollutant degradation. Despite the fact that FJH is now well known, the authors use the technique to develop a superior AOP material, which is thoroughly described and complete.

A: Thank you so much for your comments. Your positive comments will be contributed to our research. And we will check this manuscript according to your suggestion.

Q1 *The authors need to thoroughly improve the english throughout the manuscript to improve clarity.*

A: We made language polishing by a native speaker.

References

1. Yu, J., Hou, X., Hu, X., Yuan, H., Wang, J. & Chen, C. Efficient degradation of chloramphenicol by zero-valent iron microspheres and new insights in mechanisms. *Appl Catal B-Environ* **256**, 117876 (2019).
2. Ricardo, I. A., *et al.* Degradation and initial mechanism pathway of chloramphenicol by photo-Fenton process at circumneutral pH. *Chem Eeg J.* **339**, 531-538 (2018).
3. Zhang, T., *et al.* Synergistic degradation of chloramphenicol by ultrasound-enhanced nanoscale zero-valent iron/persulfate treatment. *Sep and Purif Technol.* **240**, 116575 (2020).
4. Xiang, L., *et al.* Integrating biochar, bacteria, and plants for sustainable remediation of soils contaminated with organic pollutants. *Environ Sci Technol.* **56**, 16546-16566 (2022).
5. Hodges, B. C., Cates, E. L. & Kim, J. H. Challenges and prospects of advanced oxidation water treatment processes using catalytic nanomaterials. *Nat Nanotechnol.* **13**, 642-650 (2018).
6. Shang, Y., Xu, X., Gao, B., Wang, S. & Duan, X. Single-atom catalysis in advanced oxidation processes for environmental remediation. *Chem Soc Rev.* **50**, 5281-5322 (2021).
7. Huang, J., *et al.* Fe(II) redox chemistry in the environment. *Chem Rev.* **121**, 8161-8233 (2021).
8. Wang, Z., *et al.* Is sulfate radical really generated from peroxydisulfate activated by iron (II) for environmental decontamination? *Environ Sci Technol.* **52**, 11276-11284 (2018).
9. Cai, C., Zhang, H., Zhong, X. & Hou, L. Ultrasound enhanced heterogeneous activation of peroxymonosulfate by a bimetallic Fe-Co/SBA-15 catalyst for the degradation of Orange II in water. *J Hazard Mater.* **283**, 70-79 (2015).
10. Yao, Y., *et al.* Carbothermal shock synthesis of high-entropy-alloy nanoparticles. *Science.* **359**, 1489-1494 (2018).
11. Li, Y. In situ investigation of the valence states of iron-bearing phases in chang'E-5 lunar soil using FIB, AES, and TEM-EELS techniques. *Atom Spectrosc.* **43**, 53-59 (2022).
12. Feldhoff, A., *et al.* Spin-state transition of iron in $(\text{Ba}_{0.5}\text{Sr}_{0.5})(\text{Fe}_{0.8}\text{Zn}_{0.2})\text{O}_{3-\delta}$ perovskite. *J Solid State Chem.* **182**, 2961-2971 (2009).
13. Luong, D. X., *et al.* Gram-scale bottom-up flash graphene synthesis. *Nature.* **577**, 647-651 (2020).
14. Seo, J. Y., Choi, H., Kim, C. S., Uhm, Y. R. & Sun, G. M. Determination of the magnetic structure and properties of the FeS compound by using Mössbauer spectroscopy. *J Korean Phys Soc.* **77**, 898-902 (2020).
15. Song, F., *et al.* Novel insights into the kinetics, evolved gases, and mechanisms for biomass (sugar cane residue) pyrolysis. *Environ Sci Technol.* **53**, 13495-13505 (2019).
16. Wyss, K. M., Chen, W., Beckham, J. L., Savas, P. E. & Tour, J. M. Holey and wrinkled flash graphene from mixed plastic waste. *ACS Nano.* **16**, 7804-7815 (2022).
17. Chen, J., Zhou, X., Sun, P., Zhang, Y. & Huang, C. H. Complexation enhances Cu(II)-activated peroxydisulfate: A novel activation mechanism and Cu(III) contribution. *Environ Sci Technol.* **53**, 11774-11782 (2019).
18. Nie, M., Yan, C., Li, M., Wang, X., Bi, W. & Dong, W. Degradation of chloramphenicol by

- persulfate activated by Fe²⁺ and zerovalent iron. *Chem Eeg J.* **279**, 507-515 (2015).
19. Nie, M., *et al.* Degradation of chloramphenicol using a combination system of simulated solar light, Fe²⁺ and persulfate. *Chem Eeg J.* **348**, 455-463 (2018).
 20. Guo, H., *et al.* Degradation of antibiotic chloramphenicol in water by pulsed discharge plasma combined with TiO₂/WO₃ composites: mechanism and degradation pathway. *J Hazard Mater.* **371**, 666-676 (2019).

REVIEWER COMMENTS

Reviewer #1 (Remarks to the Author):

The revision has been improved according to the comments which were carefully responded, I recommend its acceptance for publication in Nature Communications in this form.

Reviewer #2 (Remarks to the Author):

Review for "Instantaneous self-heating boost Fe-based material for advanced oxidation decontamination"

I thank the authors' for their efforts to revise the manuscript, which has been improved. However, the authors' responses did not totally address some of my concerns. Unless these comments can be appropriately addressed, I cannot recommend publication.

1. During last round of review, I suggested a possible carbothermic reduction mechanism of FeS to Fe(0). The authors argued in the rebuttal letter that FeS cannot be reduced by carbothermic reduction, but the reasons the authors listed are not convincing to me. Let's consider the below two reactions:

Decomposition: $2\text{FeS}(s) = \text{Fe}(s) + \text{S}_2(g)$ (1)

Carbothermic Reduction: $2\text{FeS}(s) + \text{C}(s) = \text{Fe}(s) + \text{CS}_2(g)$ (2)

The thermodynamic analysis of the two reactions should be done to clarify which reaction is more favorable under specific reaction temperatures.

The authors conducted an experiment that using a pyrolysis temperature of 1173 K, the FeS remains to be FeS (Fig. 4 in rebuttal letter), implying the reaction (2) doesn't happen. However, as suggested by the authors, the FJH temperature is about 1800 K. If one reaction does not happen at 1173 K, won't it be possible to happen at 1800 K? Hence their control experiment didn't tell much.

2. On the evidence of Fe(0)/FeS₂ heterostructure, the authors conducted EELS measurement, as I advised, which is appreciated. In line 160, the authors mentioned that "HAADF-STEM with EDS may infer the FeO was existed in surface of FeS." This conclusion is premature. One cannot judge whether the structure is Fe(0)/FeS₂/C, or FeS₂/Fe(0)/C, just based on the STEM image (Fig. 2f). More evidence maybe needed to confirm the proposed Fe(0)/FeS₂/C structure. This is important because then later the calculations are based on the model of Fe(0)/FeS/C.

3. Line 167, "due to the rapidly cooling at a cooling rate of 105 K/s 12." The authors cited a reference (Nature. 577, 647-651, 376 (2020)). In the later paper, the cooling rate of FJH is on the order of 104 K/s. Did the authors here achieve a cooling rate of 105 K/s? If so, temperature profile should be provided;

otherwise, this value should be corrected. And the cooling rate should be essentially dependent on the sample size where a larger sample dimension leads to slower thermal dissipation thus a slower cooling rate.

4. In last round of review, other than the Fe(0)/FeS₂ heterostructure, I requested the authors to discuss the effect of other Fe species on the catalytic performance, given that the Fe species is very complicated (FeS, Fe₇S₈, FeO, Fe₃C, and Fe₂O₃). The authors here provided controlled experiments on Fe₃C, Fe₂O₃. At a minimum, the contribution of FeS and Fe(0) should be clarified. The authors observed the Fe(0)/FeS heterostructure, so they attributed the catalytic performance to the heterostructure. This is somewhat subjective. The catalytic active sites should be better clarified.

5. Lastly, for the theoretical model, as I mentioned in #2, if the authors cannot explicitly confirm the Fe(0)/FeS/C structure, then the FeS/Fe(0)/C may also need to be considered in the model.

Other small issues:

1. No y-axis in Fig. 1e.
2. Scale bar is required for Fig. 1c.
3. Fig. 5, furnace limit at 1250 K. This is incorrect. Actually, high temperature furnace is capable of reaching up to 1800 °C or even higher.

Reviewer #3 (Remarks to the Author):

Fig 11 has issues as more peaks can be seen in the 5 minutes chromatograph than 15 minutes. Explain this and add 0-minute degradation (Control).

Reviewer #1 (Remarks to the Author):

The revision has been improved according to the comments which were carefully responded, I recommend its acceptance for publication in Nature Communications in this form.

A: We are very appreciated for your approval on our revised manuscript.

Reviewer #2 (Remarks to the Author):

Review for "Instantaneous self-heating boost Fe-based material for advanced oxidation decontamination" I thank the authors' for their efforts to revise the manuscript, which has been improved. However, the authors' responses did not totally address some of my concerns. Unless these comments can be appropriately addressed, I cannot recommend publication.

A: We are very appreciated and agree with you about your professional comments and suggestions. In order to solve some of your concerns, we added some experiments and calculation analysis, such as gas production detection, thermodynamic analysis, the control experiment of FeS and Fe⁰, and DFT calculation. We will be happy to edit the text further, based on helpful comments from you.

Q1. *During last round of review, I suggested a possible carbothermic reduction mechanism of FeS to Fe⁰. The authors argued in the rebuttal letter that FeS cannot be reduced by carbothermic reduction, but the reasons the authors listed are not convincing to me. Let's consider the below two reactions:*

Decomposition: $2\text{FeS (s)} = 2\text{Fe (s)} + \text{S}_2 \text{(g)}$ (1)

Carbothermic Reduction: $2\text{FeS (s)} + \text{C (s)} = \text{Fe (s)} + \text{CS}_2 \text{(g)}$ (2)

The thermodynamic analysis of the two reactions should be done to clarify which reaction is more favorable under specific reaction temperatures.

The authors conducted an experiment that using a pyrolysis temperature of 1173 K, the FeS remains to be FeS (Fig. 4 in rebuttal letter), implying the reaction (2) doesn't happen. However, as suggested by the authors, the FJH temperature is about 1800 K. If one reaction does not happen at 1173 K, won't it be possible to happen at 1800 K? Hence their control experiment didn't tell much.

A: Thank you for your professional advice. We are sorry for unprecise responding in the foregone rebuttal letter. In order to further solve these concerns, we added the thermodynamic analysis and measured the reacted products from FJH processes for judging the decomposition and carbothermic reduction reactions.

The thermodynamic reactions of FeS with or without carbon were calculated by FactSage software with the reaction module and the FactPS databases. The pressure of pure solids or pure liquids were 0.375 atm (according to the vacuum reaction), and the pressure of gas products was 10^{-6} atm (at the reacted begin, the pressure of S_2 and CS_2 were close to zero), and the temperature range was 300~2500 K¹. In addition, the CS_2 and S_2 gas product from FJH processes were collected by solvent condensation according to the literature².

(1) Results of thermodynamic analysis indicated that at the same temperature, the reaction of carbothermic reduction was slightly easier than that of decomposition due to the lower Gibbs free energy (**Fig. 1**). When the reacted temperature was higher than 1400 K, Gibbs free energy was less than 0 and these two reactions were all spontaneous.

(2) The CS_2 gas product from FJH process was collected by ethanol solution. GC-MS results also indicated the weak peaks of CS_2 was appeared (**Fig. 2**). S_2 was collected by CS_2 solution. Weak peak of S_2 from FJH process was detected by HPLC (**Fig. 3**).

Results above of thermodynamic analysis and experiment proved that the decomposition and carbothermic reduction reactions were co-existed in the FJH process.

The detected results of CS_2 and S_2 gas from FJH processes were added in supporting materials as **Supplementary Fig. 16 and 17**. And the corresponding explanation was as follows.

The formation reason was that Fe^0 was produced from fast breaking Fe-S bond **due to the decomposition and carbothermic reduction reactions (Fig. S16 and S17) and partial Fe^0 was sharply solidified near to FeS for forming Fe^0/FeS heterostructure due to the rapidly cooling rate (up to 541 K/s in Fig. S18)³.**

The detected method was added in supporting materials as follows.

The CS_2 and S_2 gas from FJH processes were collected by ethanol and CS_2 solvent condensation according to the literature². The CS_2 was detected by GC-MS (Agilent 5977B GC/MSD) with a HP-Wax column. The GC oven temperature was programmed

at a rate of 30 °C/min from 60 °C (held for 2 min) to 230 °C (held for 5 min). The carrier gas was helium with a flow rate of 1 mL/min, and 1 μ L sample was injected at 230 °C. S_2 concentrations was detected rapidly by HPLC at a wavelength of 254 nm, 0.8 mL/min mobile phase (the volume ratio of methanol to ultrapure water is 95:5), and the column temperature is 25 °C.

Fig. 1 The relations of ΔG and temperature in the different reaction systems.

Fig. 2 The CS₂ detection by ethanol collection liquid from FJH process.

Fig. 3 The sulfur peak were detected by HPLC from sulfur standard and Fe-C-FJH.

2. On the evidence of Fe^0/FeS heterostructure, the authors conducted EELS measurement, as I advised, which is appreciated. In line 160, the authors mentioned that “HAADF-STEM with EDS may infer the Fe^0 was existed in surface of FeS .” This conclusion is premature. One cannot judge whether the structure is $Fe^0/FeS/C$, or $FeS/Fe^0/C$, just based on the STEM image (Fig. 2f). More evidence maybe needed to confirm the proposed $Fe^0/FeS/C$ structure. This is important because then later the calculations are based on the model of $Fe^0/FeS/C$.

A: We are appreciated for your suggestion and found it was difficult to distinguish the $Fe^0/FeS/C$ or $FeS/Fe^0/C$. Therefore, according to your great suggestion in question 5, we also added the DFT calculation on the model of $FeS/Fe^0/C$. Results proved that the difference of $Fe^0/FeS/C$ and $FeS/Fe^0/C$ was negligible in adsorption energy, O-O bond of PDS, and electron transmission ability. In addition, electron transmission ability of $Fe^0/FeS/C$ and $FeS/Fe^0/C$ were notably higher than FeS/C and Fe^0/C (**Fig. 4a**). And $Fe^0/FeS/C$ and $FeS/Fe^0/C$ easily adsorbed PDS and broke O-O bond to produce radicals due to the increase of electron transmission ability (**Fig. 4b**).

Fig. 4 a-b, The optimized adsorption structures system of different materials with PDS for calculating (a) electron transmission ability and (b) the adsorption energy, O-O bond of PDS.

We deleted the inappropriate explanation in the revised manuscript about “HAADF-STEM with EDS may infer the Fe⁰ was existed in surface of FeS.” And the adsorption energy, O-O bond length of PDS, and electron transmission ability of FeS/Fe⁰/C model were added in **Supplementary Fig. 30b and c**. The corresponding explanation was added as follow.

The catalytic mechanism of nano Fe⁰/FeS heterostructure composite was further explained by the DFT calculation. We structured two species structure models (Fe⁰/FeS/C and FeS/Fe⁰/C) about Fe⁰/FeS heterostructure embedded in graphene. The adsorption energy on Fe⁰/FeS/C and FeS/Fe⁰/C were -4.169 eV and -4.135 eV.

respectively. The O-O bond length of PDS were 1.622 Å and 1.619 Å, respectively. Result indicated that compared to the adsorption systems of FeS/C or Fe⁰/C with PDS, Fe⁰/FeS heterostructure easily adsorbed PDS and broke O-O bond to produce radicals due to the increase of electron transmission ability. Electron transmission ability of Fe⁰/FeS/C and FeS/Fe⁰/C were notably higher than FeS/C and Fe⁰/C (Fig. 3d and Fig. S30c) because of electron delocalization effect that more electron can be transferred^{4, 5}. Electron delocalization triggered the formation of bidentate binuclear model which provided two passageways for electron transformation between Fe atom and PDS. The difference of Fe⁰/FeS/C and FeS/Fe⁰/C was negligible in adsorption energy, O-O bond of PDS, and electron transmission ability. All results confirmed that existent electron-rich nano Fe⁰/FeS heterostructure had a great improvement for CAP degradation.

3. Line 167, “due to the rapidly cooling at a cooling rate of 10⁵ K/s¹².” The authors cited a reference (Nature. 577, 647-651, 376 (2020)). In the later paper, the cooling rate of FJH is on the order of 10⁴ K/s. Did the authors here achieve a cooling rate of 10⁵ K/s? If so, temperature profile should be provided; otherwise, this value should be corrected. And the cooling rate should be essentially dependent on the sample size where a larger sample dimension leads to slower thermal dissipation thus a slower cooling rate.

A: Thank you for your professional advice about a cooling rate. And we agree with that the cooling rate should be essentially dependent on the sample size. Therefore, we added the practical cooling rate in FJH process for Fe-C-FJH preparation. Results found the cooling rate can reach 541 K/s in the FJH process, while that of Fe-C-PY was very slow (~hours) in pyrolysis process (**Fig. 5**). The faster cooling rate will inhibit metal agglomeration for improving material performance. According to the practical results, we corrected the cooling rate in this revised manuscript as follow.

The formation reason was that Fe⁰ was produced from fast breaking Fe-S bond **due to the decomposition and carbothermic reduction reactions (Fig. S16 and S17) and**

partial Fe^0 was sharply solidified near to FeS for forming Fe^0/FeS heterostructure due to the rapidly cooling rate (up to 541 K/s in Fig. S18)³.

Fig. 5 a-b, The temperatures were recorded in the (a) FJH and (b) pyrolysis cooling processes for Fe-C-FJH and Fe-C-PY preparation.

4. In last round of review, other than the Fe^0/FeS heterostructure, I requested the authors to discuss the effect of other Fe species on the catalytic performance, given that the Fe species is very complicated (FeS , Fe_7S_8 , Fe^0 , Fe_3C , and Fe_2O_3). The authors here provided controlled experiments on Fe_3C , Fe_2O_3 . At a minimum, the contribution of FeS and Fe^0 should be clarified. The authors observed the Fe^0/FeS heterostructure, so they attributed the catalytic performance to the heterostructure. This is somewhat subjective. The catalytic active sites should be better clarified.

A: Thank you for your suggestion. According to your suggestion, we added the controlled experiments on FeS and Fe^0 for CAP removal. Results indicated that CAP removal of FeS or Fe^0 or the mixture was lower than Fe-C-FJH (**Fig. 6**). Therefore, the specific heterostructure of Fe^0/FeS had a better improvement on CAP removal at the same content of FeS and Fe^0 .

Fig. 6 a-b, The control experiments on CAP removal of (a) FeS or Fe⁰ independently and (b) FeS or Fe⁰ or the mixture of FeS and Fe⁰ with PDS. The added Fe content was same with that of Fe-C-FJH.

5. Lastly, for the theoretical model, as I mentioned in #2, if the authors cannot explicitly confirm the Fe⁰/FeS/C structure, then the FeS/Fe⁰/C may also need to be considered in the model.

A: Thank you for your suggestion. According to your great suggestion, we structured the FeS/Fe⁰/C model for calculating the adsorption energy, O-O bond length of PDS, and electron transmission ability. Results indicated that FeS/Fe⁰/C has a similar promotion for activating PDS, compared with Fe⁰/FeS/C. Electron transmission ability (0.218 e) of FeS/Fe⁰/C was notably higher than FeS/C (0.11 e) and Fe⁰/C (0.18 e) (**Fig. 4**). Meanwhile, FeS/Fe⁰/C easily adsorbed PDS and broke O-O bond to produce radicals due to the increase of electron transmission ability. The results also confirmed that FeS/Fe⁰/C also had a great improvement for CAP degradation.

And the adsorption energy, O-O bond length of PDS, and electron transmission ability of FeS/Fe⁰/C model were added in **Supplementary Fig. 30b and c**. The corresponding explanation was added in the revised manuscript.

Other small issues:

Q1. No y-axis in Fig. 1e.

A: We have added the y-axis in Fig. 1e in the revised manuscript.

Fig. 7 Kinetic analysis of catalytic CAP degradation for explaining the role of carbon substrate. FJH with different precursors independently, hydrochar precursor, FeS precursor, and composite precursor of hydrochar and FeS.

Q2. Scale bar is required for Fig. 1c.

A: We have added the scale bar in Fig. 1c and Supplementary Fig. 2a in the revised manuscript.

Fig. 8 Lighting intensity changed in the reacted phenomenon recorded by high-speed camera.

Fig. 9 Reacted phenomenon of different time recorded by a high-speed camera in the FJH process.

Q3. Fig. 5, furnace limit at 1250 K. This is incorrect. Actually, high temperature furnace is capable of reaching up to 1800 °C or even higher.

A: Thank you for pointing this. In the original manuscript, the furnace limit at 1250 K according to literature⁶ and common pyrolysis temperature. Now, we have corrected this expression in the revised manuscript and deleted the furnace limit.

Fig. 10 Summary of nanomaterials synthesis temperatures and duration time by conventional technologies in the literatures.

Reviewer #3 (Remarks to the Author):

Fig 11 has issues as more peaks can be seen in the 5 minutes chromatograph than 15 minutes. Explain this and add 0-minute degradation (Control).

A: Thank you so much for pointing that more peaks can be seen in the 5 minutes chromatograph than 15 minutes and your suggestions. We redid the experiments at different degradation time and measured the intermediate products by GC-MS.

By measuring the former ethyl acetate used for extracting organics from reacted liquid, the obvious peaks of 10.293, 13.371, and 14.398 min were appeared for the degradation time of 5 and 55 minutes, while these peaks were not appeared for the degradation time of 15 minutes (**Fig 11 and Table 1**). The reason was due to the background of ethyl acetate from different batch or other impurity substance. Before the measurement of GC-MS, ethyl acetate was used to extract organics from reacted liquid.

Therefore, we redid the experiments and removed the effect of ethyl acetate. Results indicated there are also seven intermediate products that were same with the former results except the peak of 9.137 min. The peak area of CAP in chromatograph were decreased with increasing time (**Fig 12 and Table 2**). In addition, we added the 0 minutes chromatograph. We rapidly added the catalytic material and PDS into CAP solution, and immediately took reacted solution for GCMS measurement. Results found partial CAP can be also quickly degraded (**Fig. 12**). The GC-MS results were readded as the **Supplementary Fig. 26 and Supplementary Fig. 28**.

Fig. 11 The chromatograph of only ethyl acetate used for extracting organics from reacted liquid.

Table 1 The peak components of only ethyl acetate used for extracting organics from reacted liquid at different degradation time.

Time (min)	Components
10.293	Dibutyl phthalate (C ₁₆ H ₂₂ O ₄)
13.371	2,2-Methylenebis (6-(1,1-dimethylethyl)-4-methyl-phenol (C ₂₃ H ₃₂ O ₂)
14.394	1,1,3,3,5,5,7,7,9,9,11,11,13,13,15,15-hexadecamethyl-octasiloxane (C ₁₆ H ₅₀ O ₇ Si ₈)

Fig. 12 Intermediate products of CAP degradation by Fe-C-FJH at different time were measured by GC-MS.

Table 2 The peak area of intermediate products of CAP degradation by Fe-C-FJH were measured from GC-MS.

Time (min)	Components				
		0 min	5 min	15 min	55 min
4.701	2,2-dichloro-N-(3-methylbutyl) acetamide (C ₇ H ₁₃ Cl ₂ NO)	1162741	1457355	1396107	1006593
5.857	3-Hydroxy-4-nitrobenzaldehyde (C ₇ H ₅ NO ₄)	-	-	-	54483
7.094	4-Nitrophenol (C ₆ H ₅ NO ₃)	-	-	-	79895
7.318	4-Nitrobenzoic acid (C ₇ H ₅ NO ₄)	-	241786	556145	857754
9.137	4-Nitrophenylglyoxylic acid (C ₈ H ₅ NO ₅)	-	440575	744960	1060515
12.596	2,2-dichloro-N-(1-(4-nitrobenzoyl) vinyl) acetamide (C ₁₁ H ₈ Cl ₂ N ₂ O ₄)	1074324	3904691	5757322	5437289
13.163	4-nitro-Benzaldehyde (C ₇ H ₅ NO ₃)	2816010	4796613	4963104	3446899
16.161	Chloramphenicol (C ₁₁ H ₁₂ Cl ₂ N ₂ O ₅)	29052387	12920902	8639573	3293566

References

1. Zheng, Z., You, Y., Guo, J., Li, G., You, Z. & Lv, X. Pyrolysis Behavior of Pyrite under a CO-H₂ Atmosphere. *ACS Omega*. **7**, 29116-29124 (2022).
2. Deng, B., Luong, D. X., Wang, Z., Kittrell, C., McHugh, E. A. & Tour, J. M. Urban mining by flash Joule heating. *Nat Commun*. **12**, 5794 (2021).
3. Luong, D. X., *et al.* Gram-scale bottom-up flash graphene synthesis. *Nature*. **577**, 647-651 (2020).
4. Guo, Z. Y., *et al.* Electron delocalization triggers nonradical Fenton-like catalysis over spinel oxides. *Proc Natl Acad Sci USA*. **119**, e2201607119 (2022).
5. Zhang, P., Yang, Y., Duan, X., Liu, Y. & Wang, S. Density functional theory calculations for insight into the heterocatalyst reactivity and mechanism in persulfate-based advanced oxidation reactions. *ACS Catal*. **11**, 11129-11159 (2021).
6. Yao, Y., *et al.* High temperature shockwave stabilized single atoms. *Nat Nanotechnol*. **14**, 851-857 (2019).

REVIEWERS' COMMENTS

Reviewer #2 (Remarks to the Author):

Review for "Instantaneous self-heating boost Fe-based material for advanced oxidation decontamination"

The authors have clarified most of my concerns. I have one last question on the theoretical model. As shown in Fig. 3d and Fig. S30, the optimized structures of PDS on FeS/C and FeS/Fe/C are rather different, i.e., 1O bonding on FeS/C and 2O bonding on FeS/Fe/C. This pertains to the absorbed configuration of PDS on Fe/C and Fe/FeS/C. What is the convergence conditions used for the structure optimization?

Reviewer #3 (Remarks to the Author):

The query has been answered; however, making a proper degradation pathway is advisable. The proposed pathways (SI) are not proper; also, add chemical structure with Table 2. Furthermore, take the help of an expert if required.

Reviewer #2 (Remarks to the Author):

Review for "Instantaneous self-heating boost Fe-based material for advanced oxidation decontamination". The authors have clarified most of my concerns. I have one last question on the theoretical model. As shown in Fig. 3d and Fig. S30, the optimized structures of PDS on FeS/C and FeS/Fe/C are rather different, i.e., 1O bonding on FeS/C and 2O bonding on FeS/Fe/C. This pertains to the absorbed configuration of PDS on Fe/C and Fe/FeS/C. What is the convergence conditions used for the structure optimization?

A: Thank you for professional suggestion. The convergence tolerances of energy and force in all structure were set to 1.0×10^{-4} eV atom⁻¹ and 1.0×10^{-2} eV Å⁻¹, respectively. And the modification of van der Waals force was also considered. When the absorbed configuration of PDS on FeS/C, Fe/C, Fe/FeS/C, and FeS/Fe/C were structured, we have considered the 2O bonding on all configuration. After the optimization of these absorbed configuration, 1O bonding on FeS/C and Fe/C, 2O bonding on Fe/FeS/C and FeS/Fe/C were acquired. The reason was that electron delocalization of heterostructure composite triggered the formation of bidentate binuclear model which provided two passageways for more electron transformation between Fe atom and PDS.

Reviewer #3 (Remarks to the Author):

The query has been answered; however, making a proper degradation pathway is advisable. The proposed pathways (SI) are not proper; also, add chemical structure with Table 2. Furthermore, take the help of an expert if required.

A: We are so happy for solving your query and thank you for professional suggestion. The chemical structure of intermediate products was added in the Table 1. According to the intermediates produced at different time and the observations from previous reports^{1, 2, 3, 4, 5, 6}, we also provided further clarification on the degradation pathway of the initial pollutant to intermediates and final products in the Figure 1. The CAP degradation pathway was explained in the revised manuscript, and the corresponding sentences were follows.

The degradation intermediate products from the system of CAP/PDS/Fe-C-FJH at different degradation time were shown in **Supplementary Figure 26 and 27**. Consistent with the observations from previous reports^{40, 41, 42, 43, 44, 45}, the main degradation reactions of CAP degradation included oxidation, dehydrogenation, cleavage of the amide group, and hydroxylation for producing small molecular components. Importantly, these small molecular components were further mineralized into inorganic ions (such as NO_3^- and Cl^-) by dehalogenation, denitration, and oxidation reaction (**Supplementary Table 6**). Therefore, the proposed degradation pathway was shown in **Supplementary Figure 28**.

Figure 1. Proposed CAP degradation pathways from the system of CAP/PDS/Fe-C-FJH.

Table 1 The peak area of intermediate products of CAP degradation by Fe-C-FJH were measured from GC-MS.

Time (min)	Components	Chemical structure	Peak Area			
			0 min	5 min	15 min	55 min
4.701	2,2-dichloro-N-(3-methylbutyl)acetamide ($C_7H_{13}Cl_2NO$)		1162741	1457355	1396107	1006593
5.857	3-Hydroxy-4-nitrobenzaldehyde ($C_7H_5NO_4$)		-	-	-	54483
7.094	4-Nitrophenol ($C_6H_5NO_3$)		-	-	-	79895
7.318	4-Nitrobenzoic acid ($C_7H_5NO_4$)		-	241786	556145	857754
9.137	4-Nitrophenylglyoxylic acid ($C_8H_5NO_5$)		-	440575	744960	1060515
12.596	2,2-dichloro-N-(1-(4-nitrobenzoyl)vinyl)acetamide ($C_{11}H_8Cl_2N_2O_4$)		1074324	3904691	5757322	5437289
13.163	4-nitro-Benzaldehyde ($C_7H_5NO_3$)		2816010	4796613	4963104	3446899
16.161	Chloramphenicol ($C_{11}H_{12}Cl_2N_2O_5$)		29052387	12920902	8639573	3293566

References

1. Yu, J., Hou, X., Hu, X., Yuan, H., Wang, J. & Chen, C. Efficient degradation of chloramphenicol by zero-valent iron microspheres and new insights in mechanisms. *Appl Catal B-Environ* **256**, 117876 (2019).
2. Nie, M., Yan, C., Li, M., Wang, X., Bi, W. & Dong, W. Degradation of chloramphenicol by persulfate activated by Fe²⁺ and zerovalent iron. *Chem Eeg J* **279**, 507-515 (2015).
3. Nie, M., *et al.* Degradation of chloramphenicol using a combination system of simulated solar light, Fe²⁺ and persulfate. *Chem Eng J.* **348**, 455-463 (2018).
4. Guo, H., *et al.* Degradation of antibiotic chloramphenicol in water by pulsed discharge plasma combined with TiO₂/WO₃ composites: mechanism and degradation pathway. *J Hazard Mater.* **371**, 666-676 (2019).
5. Zhan, H., Zhou, R., Wang, P. & Zhou, Q. Selective hydroxyl generation for efficient pollutant degradation by electronic structure modulation at Fe sites. *Proc Natl Acad Sci U S A.* **120**, e2305378120 (2023).
6. Zhou, Q., Song, C., Wang, P., Zhao, Z., Li, Y. & Zhan, S. Generating dual-active species by triple-atom sites through peroxymonosulfate activation for treating micropollutants in complex water. *Proc Natl Acad Sci U S A.* **120**, e2300085120 (2023).